



# Characterizing the Arctic absorbing aerosol with multi-instrument observations

Eija Asmi[1], John Backman[1], Henri Servomaa[1], Aki Virkkula[1], Maria Gini[2], Kostas Eleftheriadis[2], Thomas Müller[3], Sho Ohata[4,5], Yutaka Kondo[6], and Antti Hyvärinen[1]

[1]Finnish Meteorological Institute, Helsinki, Finland
[2]ERL Institute of Nuclear and Radiological Science & Technology, NCRS Demokritos, Attiki, Greece
[3]Leibniz Institute for Tropospheric Research e.V. (TROPOS), Leipzig, Germany
[4]Institute for Space–Earth Environmental Research, Nagoya University, Nagoya, Aichi, Japan
[5]Institute for Advanced Research, Nagoya University, Nagoya, Aichi, Japan
[6]National Institute of Polar Research, Tachikawa, Japan

**Correspondence:** Eija Asmi (eija.asmi@fmi.fi)

**Abstract.** The Arctic absorbing aerosols have a high potential to accelerate global warming. Accurate and sensitive measurements of their concentrations, variability and atmospheric mixing are needed. Filter-based aerosol light absorption measurement methods are the most widely applied in the Arctic. Those will be the focus of this study. Aerosol light absorption was measured during one month field campaign in June–July 2019 at the Pallas Global Atmospheric Watch (GAW) station in north-

ern Finland. The campaign provided a real-world test for different absorption measurement techniques supporting the goals of the EMPIR BC metrology project in developing aerosol absorption standard and reference methods. Very low aerosol concentrations prevailed during the campaign which imposed a challenge for the instruments detection. In this study we compare the results from five filter-based absorption techniques: Aethalometer models AE31 and AE33, Particle Soot Absorption Photometer (PSAP), Multi Angle Absorption Photometer (MAAP) and Continuous Soot Monitoring System (COSMOS), and from

one indirect method called Extinction Minus Scattering (EMS). The sensitivity of the filter-based techniques was adequate to measure aerosol light absorption coefficients down to around 0.05 $\mathrm{Mm}^{-1}$ levels. The average value measured during the campaign using MAAP was 0.09 $\mathrm{Mm}^{-1}$ (at wavelength of 637 nm). When data were averaged for >1h, an agreement of around 20% was obtained between instruments. COSMOS measured systematically the lowest absorption coefficient values, which was expected due to the sample pre-treatment in COSMOS inlet. PSAP showed the best linear correlation with MAAP ($R^2$ =

0.85), followed by AE33 and COSMOS ($R^2$ = 0.84). The noisy data from AE31 resulted in a slightly lower, yet a significant, correlation with MAAP ($R^2$ = 0.46). In contrast to the filter-based techniques, the sensitivity of the indirect EMS method to measure aerosol absorption was not adequate at such low concentrations levels. An absorption coefficient on the order of >1 $\mathrm{Mm}^{-1}$ was estimated as the lowest limit, to reliably distinguish the signal from the noise. Throughout the campaign the aerosol was highly scattering with an average single-scattering albedo of 0.97. Two different air-mass origins could be identified: the

north-east and from the north-west. The north-eastern air masses contained higher fraction of thickly coated light absorbing particles than the westerly air masses. Aerosol scattering, absorption and the particle coating thickness increased on the last ten days of the campaign during the north-eastern air flow. The simultaneous changes in aerosol source region, mixing state,



concentration and particle optical size were reflected in the instruments' response in a complex way. The observed decrease in aerosol size suggested additional activation of secondary particle formation mechanisms. The results demonstrate the challenges encountered in the Arctic absorbing aerosol measurements. The applicability and uncertainties of different techniques are discussed and new knowledge on the absorbing aerosol characteristics in summer Arctic air masses reference to the source region is provided.

# 1 Introduction

Black Smoke aerosols have been measured since the early 50's. The development of a filter-based measurement method begun with an experiment by Rosen et al. (1978). The Raman spectral measurements confirmed that the light attenuation is proportional to the graphic soot content on a filter. After this discovery the development continued by Hansen et al. (1982, 1984), and today, the various filter-based techniques are commonly used in aerosol absorption measurements. The filter-based methods are sensitive, simple and robust, and therefore widely applicable.

Meanwhile, it has become evident that the filter-based methods are prone to several filter artifacts. These include the dependence of light attenuation on the filter tape loading and the interference of aerosol light scattering with the absorption measurement (Müller et al., 2011). Aerosol size affects the penetration depth in a filter adding another size dependent measurement artifact (Kondo et al., 2009; Nakayama et al., 2010). Additional sources of uncertainties are the variations in filter spot size and the non-idealities of light source (Bond et al., 1999). Various algorithms to correct for these artifacts have been developed (Bond et al., 1999; Weingartner et al., 2003; Arnott et al., 2005; Schmid et al., 2006; Virkkula et al., 2007; Nakayama et al., 2010; Ogren , 2010; Virkkula , 2010; Collaud Coen et al., 2010). The diverse use of these algorithms complicates a direct comparison of aerosol absorption values from different studies.

Alternative absorption measurement methods exist. They are less prone to measurement artifacts and have been used for development of algorithms to remedy the uncertainties associated with the filter-based techniques. Photoacoustic techniques have the advantage to measure particle absorption in their natural atmospheric state suspended in air (Arnott et al., 1999). However, they suffer from artifacts related to the gas composition and are less robust and sensitive than the filter-based techniques. An individual particle analysis with a laser-induced incandescence (LII) technique is to date the most accurate and sensitive method to measure the absorbing mass content, the so called refractory Black Carbon (rBC) mass, of the aerosol. The existing LII techniques are expensive and complex, and converting the rBC signal to atmospheric absorption is not straightforward (Schulz et al., 2006; Schwarz et al., 2006). The measured aerosol light absorption is frequently reported as equivalent Black Carbon (eBC) mass [in units: ng m$^{-3}$] which relies on a specific wavelength dependent mass absorption cross section (MAC) coefficient (Petzold et al., 2013). A simultaneous measurement of the aerosol extinction and scattering is yet another alternative that allows to derive the aerosol absorption indirectly (Strawa et al., 2003; Virkkula et al., 2005). A review of methods with their common pros and cons is provided by Moosmuller et al. (2009).

The different methods to measure aerosol light absorption have been compared and verified in previous laboratory (Saathoff et al., 2003; Slowik et al., 2007; Müller et al., 2011) and field (Reid et al., 1998; Schmid et al., 2006; Kanaya et al., 2008;



Kondo et al., 2011; Backman et al., 2017) campaigns. The campaigns have focused on characterizing uncertainties of the different absorption techniques and examined their response to varying absorbing aerosol sources. Reid et al. (1998) measured Brasilian biomass burning aerosol using six different techniques, concluding about 20% convergence between them. Kanaya et al. (2008) found an overall good agreement between the results of different instruments, but the discrepancies increased at high Organic Carbon (OC) content. Schmid et al. (2006) measured the Amazon biomass burning aerosol using various methods and estimated 15% and 20% accuracy, for PSAP and Aethalometer measurement, respectively. Better agreement, in terms of eBC mass, can be expected when solely non-volatile absorbing particle is analyzed avoiding any artifacts from volatile light scattering particles (Kondo et al., 2011). However, Müller et al. (2011) showed that already the variability between the absorption instruments of the same model can be up to 30%.

Accuracy of the aerosol absorption measurement methods needs to be improved to reduce the uncertainties associated with their climate impacts. Absorbing aerosol has an accelerating impact on the global temperature rise, which is further intensified over the Polar Regions due to the regional strong climate feedbacks. The aerosol light absorption, and its spatial and temporal variability, are therefore of specific concern in the Arctic. Absorption measurements in the Arctic require sensitive and robust techniques. Recently, Backman et al. (2017); Schmeisser et al. (2018) found significant spatial differences in aerosol light absorption seasonal characteristics in the Arctic. All long-term aerosol absorption data series from the Arctic are measured using filter-based methods. Backman et al. (2017) used the co-located measurements to construct a homogeneous dataset for multiple Arctic sites, but the instruments were never applied all in parallel. Such a parallel comparison in the Arctic would assist to estimate the uncertainties associated to these measurements.

The absorbing aerosol climate impact depends on both the absorbing aerosol mass and its atmospheric mixing state (Jacobson, 2001). The effect of the absorbing aerosol mixing state on spring time Arctic aerosol radiative forcing was recently estimated by Zanatta et al. (2018). However, the uncertainties in absorbing aerosol measurements techniques and in data reporting hamper to reliably estimate the absorbing aerosols' radiative forcing in the Arctic (Slowik et al., 2007; Zanatta et al., 2018).

The EMPIR BC project develops metrology for light absorption by atmospheric aerosols. It aims at finding standard reference materials that mimic atmospheric absorbing aerosol and a traceable, primary method to determine the aerosol absorption coefficients. An additional goal of the EMPIR BC project is to develop a validated transfer standard for field calibrations. The Pallas EMPIR BC campaign was the first field campaign in the project. The goal was to test the sensitivity of available absorption methods, focusing on the filter-based techniques, and to analyze changes in instruments response occurring over time and with changing aerosol characteristics. The results of the campaign adds to the knowledge about the Arctic summertime absorbing aerosol sources, concentrations and mixing state.

## 2 Methodology

### 2.1 The Pallas site description

The Pallas atmosphere ecosystem supersite is located in the northern Finland inside the Arctic. It is part of the Pallas-Sodankylä Global Atmospheric Watch (GAW) station and contributes to various national and international networks and programmes. Im-





portant in this context is the Aerosols, Clouds, and Trace gases Research InfraStructure (ACTRIS) to which Pallas provides

quality controlled and continuous data on aerosol number, size and optical properties. The main station for aerosol measurements at Pallas is on top of the Sammaltunturi fell (67° 58 N', 24° 07' E, 560 m a.g.l.) where also the EMPIR BC field campaign was organized. A detailed description of the site, its surroundings and on-going measurement programmes is available by Hatakka et al. (2003); Lohila et al. (2015).

## 2.2   EMPIR BC field campaign

The EMPIR BC field campaign took place during the Nordic summer, between 19.6 – 17.7.2019. The objective was to test different absorption measurement methods in the field to better understand the challenges related with the demanding Arctic aerosol absorption measurements. The absorbing aerosol concentrations were very low, as is typical for the Polar Regions. The specific aims were (1) to estimate the sensitivity of different absorption techniques and (2) to analyse how the instruments responded to changes in aerosol characteristics.

A summary of the instrumentation used with corresponding settings during the campaign are presented in Table 1. Each instruments operational principle and respective data corrections are presented in detail in the following section. The bulk of the instruments were connected to a common inlet which was equipped with a Particulate Matter (PM) $10\mu m$ cut-size aerosol inlet head and a nafion permapure model MD-700-48 aerosol drier. The relative humidity (RH) of the sample in the entrance of the inlet was monitored to remain <40% throughout the campaign. The total flow of 18.3 LPM was divided for the instruments

via a self-made flow-divider that consisted of six cylinder symmetric exit tubes. One of the exits was further divided using a TSI laminar flow-divider into two flows, one for each extinction monitor (CAPSex and CAPSssa, see Table 1). In addition, two instruments (SP2 and AE31, see Table 1) were connected to a slightly heated, total aerosol inlet about 3 m apart from the PM10 line. The AE31 is measuring in this inlet year-round and the SP2 has a limited measurement size range for which the inlet cut-size does not affect the result in cloud-free conditions.

## 2.3   Measured and derived aerosol optical properties

Table A1 summarizes the quantities that frequently appear in this manuscript text. All the filter-based instruments target to achieve aerosol light absorption coefficients $\sigma_{AP,\lambda}$ at instrument specific wavelengths $\lambda$, which are acquired from the measured light attenuation using signal post-processing. When referring to a corrected absorption coefficient measured using a particular technique, a notation $\sigma_{\mathrm{INST},\lambda}$, where INST is an abbreviation of the technique, is used. When referring to an aerosol light

absorption coefficient value directly reported by the instrument, $\sigma_{0,\lambda}$ is used instead.

A typical measure of aerosol "brightness" is the ratio of aerosol scattering $\sigma_{SP,\lambda}$ to aerosol extinction $\sigma_{EP,\lambda}$, a parameter called single-scattering albedo

$$\omega_{0,\lambda} = \frac{\sigma_{SP,\lambda}}{\sigma_{EP,\lambda}} = \frac{\sigma_{SP,\lambda}}{\sigma_{SP,\lambda} + \sigma_{AP,\lambda}}, \tag{1}$$





which is of great significance when assessing the radiative forcing of the aerosols. The scattering wavelength dependence is
described as

$$\sigma_{SP,\lambda_1}/\sigma_{SP,\lambda_2} = (\lambda_1/\lambda_2)^{-\alpha_{SP,\lambda}}, \tag{2}$$

where $\alpha_{SP,\lambda}$ is called the Ångström exponent of scattering and is related with the aerosol optical size. A similar wavelength
dependent parameter, $\alpha_{AP,\lambda}$, is defined for aerosol absorption.

Interpolation of $\sigma_{SP,\lambda}$ to another wavelength $\lambda$ was done by utilizing the Ångström exponent at the nearest available wave-
lengths (Anderson and Ogren, 1998). Interpolation of $\sigma_{AP,\lambda}$ to another wavelength $\lambda$ was done by assuming a simple relation
between the wavelengths and variation in $\alpha_{AP,\lambda}$ was not accounted for due to high noise in data.

### 2.4   Instruments

Data from five filter-based absorption photometers, two instruments that measure aerosol scattering, two instruments that
measure aerosol extinction and one instrument that measure refractory BC are used in this paper. The data were corrected with
the best practices considered for each particular instrument independently, following the global guidelines, literature citations
and earlier work done at the station. Additionally, particle number and meteorological data from the station were utilized in the
analysis.

The flow rate of each instrument was measured at the beginning and at the end of the campaign with a Gilian flow calibrator
(volumetric flow rate), and converted to standard (STP) conditions (0°C, 1013hPa) after which the flow correction based on
the equation 5 in Bond et al. (1999) was applied. Instrument flow rates are shown in Table 1.

#### 2.4.1   AE31

Aethalometer model AE31 (Magee Scientific Inc.) is part of the permanent installation at Pallas site since year 2005 (Li-
havainen et al., 2015). It measures the aerosol absorption coefficient at seven wavelengths: 370 nm, 470 nm, 520 nm, 590 nm,
660 nm, 880 nm and 950 nm. The measurement principle is based on the observed light attenuation caused by the particles that
are continuously collected on a filter tape (Hansen et al., 1982, 1984). The aerosol attenuation coefficient is then calculated as

$$\sigma_{0,\lambda} = \frac{A}{Q*100} * \frac{\Delta ATN}{\Delta t}, \tag{3}$$

where A is the filter spot size, Q is the flow rate and $\Delta ATN$ is the measured change in the attenuation during the time interval
$\Delta t$. AE31 changed the filter spot automatically when a pre-set limit value of ATN = 60 was reached. The instrument reports
data in eBC mass concentration which is simply the measured aerosol absorption coefficient corrected with a wavelength
dependent specific attenuation (MAC).

The AE31 data measured at Pallas was corrected for the multiple scattering of light by filter fibers by dividing $\sigma_{0,\lambda}$ with
a multiple scattering enhancement factor, $C_0 = 3.5$, which is selected according to the global recommendation of the Global
Atmospheric Watch's World Calibrations Centre for Aerosol Physics (GAWReport No. 227; http://wmo-gaw-wcc-aerosol-
physics.org/wmo-gaw-reports.html), and is also very close to the $C_0$ factor for the Arctic given by Backman et al. (2017). The



filter loading artifact was corrected using the method by Virkkula et al. (2007, 2015), and the Pallas station specific correction

factor $k = 0.0038$ (Backman et al., 2017). Note that this correction is a loading correction only, unlike the algorithms of Arnott

et al. (2005) and Collaud Coen et al. (2010) in which a fraction of scattering coefficient is subtracted from $\sigma_{0,\lambda}$.

### 2.4.2   AE33

An updated version of the AE31 is the dual-spot aethalometer model AE33 (Drinovec et al., 2015). The instrument reports

an aerosol light absorption coefficient based on the measured attenuation on two parallel filter spots with different particle

loadings. It applies a real-time loading effect compensation algorithm that is essentially based on the work by Virkkula et al.

(2007). The Pallas AE33 uses an internal multiple scattering correction factor $C_0 = 1.57$ (Drinovec et al., 2015). This was

corrected to a value $C_0 = 3.5$ in order to comply with the global recommendation (GAWReport No. 227; http://wmo-gaw-wcc-

aerosol-physics.org/wmo-gaw-reports.html). However, no clear consensus or published recommendation for the AE33 specific

global scattering correction factor $C_0$ yet exist. The AE33 at Pallas was programmed to change the filter spot automatically

every 24h.

### 2.4.3   MAAP

The Multi Angle Absorption Photometer (MAAP) model 5012 (Thermo Scientific) has been frequently used as an absorption

reference for the filter-based absorption instrument techniques (Müller et al., 2011). It internally corrects for the scattering

artifact by using a simultaneous back-scattering measurement of the filter tape at multiple angles. In general, the data need

very little post-processing. MAAP measures absorption at a wavelength of 637 nm. A wavelength shift from the nominal

value was reported by Müller et al. (2011), and requires a correction with a multiplier 1.05. This correction was applied also

here. Pallas MAAP was set to report eBC directly at STP conditions and no further corrections were thus applied. In polluted

environments the MAAP internal data averaging procedure can lead to an artifact that needs to be corrected as suggested by

Hyvärinen et al. (2013).

### 2.4.4   PSAP

The Particle Soot Absorption Photometer (PSAP; Radiance Research) measures aerosol light attenuation at wavelengths 467,

530, and 660 nm (Bond et al., 1999). The PSAP, in contrast to the other filter-based techniques used, requires a manual filter

spot change. This was done when the transmittance reported by the instrument decreased from the initial value of 1.0 to a range

of 0.8–0.7. The flow rate of PSAP was set at 1 LPM.

PSAP records the signal, reference and dark count data at 4s time resolution, which was hourly averaged to calculate the

absorption coefficients. The data were corrected with the measured filter spot size and flow rate as suggested by Bond et al.

(1999); Ogren  (2010). An average of five spot sizes was determined to be A = 18.63 $mm^2$. The volumetric flow rate was

measured at 15 different adjusted flow rate settings at the beginning and at the end of the campaign. The results were converted

to standard flow rate and a linear fit was made to the data, resulting in a flow correction factor of 1.12.





The obtained aerosol absorption coefficient was corrected for the filter-tape loading and scattering artifacts using the correction scheme by Virkkula (2010)

$$\sigma_{\text{PSAP},\lambda} = (k0 + k1(h0 + h1 * \omega_{0,\lambda})ln(Tr_\lambda))\sigma_{0,\lambda} - s * \sigma_{SP,\lambda}, \qquad (4)$$

where k0, k1, h0, h1 and s are wavelength dependent constants given by Virkkula (2010). $Tr_\lambda$ is the transmittance measured
by PSAP at a wavelength $\lambda$ and $\sigma_{SP,\lambda}$ are the corresponding scattering coefficients. The scattering coefficients were measured
with a nephelometer (TSI Inc. model 3563) and interpolated to the three PSAP wavelengths using the calculated Ångström
exponent values $\alpha_{SP,\lambda}$. The single scattering albedo $\omega_{0,\lambda}$ in Equation 4 was iterated until no significant change in $\sigma_{\text{PSAP},\lambda}$ was
observed. At large values of $\omega_{0,\lambda}$ such as here, this correction scheme approaches the widely applied Bond-Ogren correction
scheme (Bond et al., 1999; Ogren , 2010). However, it should be noted that at high $\omega_{0,\lambda}$ values the $\alpha_{AP,\lambda}$ becomes uncertain
and should be interpreted with caution (Backman et al., 2014).

### 2.4.5 COSMOS

The Continuous soot monitoring system (COSMOS) measures light attenuation at a wavelength of 565 nm. The measurement
principle is similar to other filter-based absorption photometers. However, the sample is exposed to a pre-treatment (Miyazaki
et al., 2008; Kondo et al., 2009). In the COSMOS inlet the volatile non-refractory aerosol components are removed by heating
the sample to 300°C. The inlet characteristics are presented by Kondo et al. (2009). The COSMOS mechanical and optical
design with the determined instrument detection limit and measurement uncertainties are presented by Miyazaki et al. (2008).
Due to efficient elimination of the artifacts from aerosol scattering this method is typically found in good agreement with the
thermal-optical and the laser-induced incandescence techniques (Kondo et al., 2009, 2011), and not directly comparable to
other filter-based absorption measurements. Each COSMOS is calibrated against a standard COSMOS instrument using am-
bient absorbing aerosol within an accuracy of about 5%. The standard COSMOS, in turn, is calibrated by SP2 using ambient
absorbing aerosol and applying an aerosol specific MAC. Detailed comparison of COSMOS and SP2 measurements at several
sites in Asia and the Arctic have demonstrated that the overall accuracy in the absorbing aerosol mass concentration measure-
ment is about 10% (Ohata et al., 2019). The stability of MAC is explained by the elimination of the artifacts from aerosol
scattering. At Pallas, COSMOS was operated at 0.7 LPM flow rate (STD) and the data were saved every 1-min.

### 205 2.4.6 SP2

The Single particle soot photometer (SP2, Droplet Measurement Technologies Inc.) measures refractory Black Carbon (rBC)
mass in particles >70 nm in diameter (Schwarz et al., 2006). The measurement principle is based on a laser-induced in-
candescence where the particle is heated up to the point of incandescence which is picked up by the instruments detectors
(Stephens et al., 2003). The incandescence signal is proportional to the mass of the refractory black carbon which is calcu-
lated particle-by-particle to obtain the rBC mass concentration (Lim et al., 2014). This technique is very sensitive but does not
measure particle light absorption as such, and therefore, a direct comparison with other absorption measurement techniques is





not straightforward. SP2 data describe the mixing characteristics of the atmospheric light absorbing aerosol which has a strong impact on their direct radiative effect.

### 2.4.7 CAPS

CAPS PMex (CAPSex, Aerodyne Research Inc.) instrument measures total light extinction by aerosol particles ($\sigma_{EP,\lambda}$) utilizing a cavity attenuated phase shift principle (Kebabian et al., 2007; Massoli et al., 2010; Petzold et al., 2013; Perim de Faria et al. , 2017). An updated model of CAPSex is the CAPS PMssa (CAPSssa, Aerodyne Research Inc.). CAPSssa additionally measures the aerosol light scattering allowing the single scattering albedo to be determined with a single instrument (Onasch et al., 2015). The scattering measurement technique is similar to an integrating nephelometer, and utilizes a Lambertian inte-

grating sphere in the sample cell. The aerosol light scattering measurement by CAPSssa is affected by background, truncation and light-source related uncertainties for which calibration is needed.

Aerosol extinction measurement with CAPS is nearly a calibration free technique as long as frequent baseline measurements are performed. A potential source of systematic bias is the geometry correction factor. This is generally a stable constant but has been shown to vary between instruments of even the same model (Petzold et al., 2013; Onasch et al., 2015). An accurate

aerosol extinction measurement thus requires calibration against a calibrated scattering instrument, generally a nephelometer.

CAPSex and CAPSssa that were operated at Pallas both measure at a wavelength of 630 nm. They were calibrated in the beginning, middle and end of the campaign using ammonium sulfate aerosol generated with an atomizer.

### 2.4.8 Nephelometer

Aerosol light scattering is continuously monitored at Pallas Sammaltunturi site with an integrating nephelometer (TSI3; TSI,

model 3563) (Anderson and Ogren, 1998; Heintzenberg et al., 2006). It measures aerosol total scattering and back-scattering fraction at three wavelengths: 450 nm, 550 nm and 700 nm. Nephelometer data were corrected for truncation as suggested by Anderson and Ogren (1998) and converted to standard atmospheric conditions (STP).

During the EMPIR campaign the aerosol light scattering was also measured with an Aurora integrating polar nephelometer (AUR4; Ecotech, model 4000) at two angles: 90 and 180. In this setup, the Aurora nephelometer measures the total scattering

(180) and back-scattering (90) of the aerosol in similar manner as the TSI nephelometer. The Aurora 4000 measures scattering at wavelengths of 450 nm, 525 nm, and 635 nm. Data were corrected for truncation based on Müller et al. (2011) and converted to STP.

A zero check was performed daily for both nephelometers and they were calibrated with $CO_2$ gas in the beginning and in the end of the campaign.

### 240 2.4.9 Extinction Minus Scattering (EMS)

An indirect technique to determine the aerosol light absorption is based on separately measured aerosol extinction and aerosol scattering (Strawa et al., 2003; Virkkula et al., 2005). This extinction minus scattering (EMS) -method relies on those aerosol





optical properties that can be accurately determined using existing techniques. It is also traceable to SI units. The EMS method avoids the artifacts encountered with filter-based techniques.

In Pallas the aerosol light scattering was measured with two integrating nephelometers and the extinction with CAPSex and CAPSssa instruments. Two instrument "pairs" were formed: (1) Aurora 4000 polar nephelometer and CAPSssa (both were part of campaign instrumentation) and (2) CAPSex and the TSI nephelometer (permanent instrumentation at site). These methods here are referred to as $EMS_1$ and $EMS_2$, respectively. The CAPSssa scattering and extinction measurement alone was also used to determine aerosol absorption, which is here referred to as method $EMS_3$.

In the beginning, middle and end of the campaign the CAPS instruments data were calibrated against the nephelometers. Purely scattering ammonium sulphate aerosol was produced with an atomizer (TOPAS, model ATM230). The aerosol losses in the sampling lines and in the instruments are size dependent and the $\alpha_{SP,\lambda}$ reflects the optical size of the aerosol. In Pallas summer atmosphere a typical value of $\alpha_{SP,\lambda} = 1.5 - 1.7$. The ammonium sulphate calibrations were performed at this $\alpha_{SP,\lambda}$ range so that the calibrations will be valid for the ambient aerosol as well. A summary of the three calibrations proposed

multipliers of 1.21 ($EMS_1$) and 1.04 ($EMS_2$) for CAPS data (Figure 1). These correction factors were applied to all CAPS data in this manuscript. The different correction factors for the CAPSex and CAPSssa could be explained by the different individual geometry correction factors, and to a lesser extent, by the discrepancies in inlet tubing sizes and flow rates. The two CAPS were having identical flow rates and inlet settings, and the two nephelometers (reference points) had a small difference in flow rates and in inlet tubing sizes. Major care was taken in measurement system construction to provide similar sample intake to

all instruments.

### 2.4.10   Auxiliary measurements

The particle total number concentration at >10 nm size was measured with a Condensation Particle Counter (CPC; TSI model 3772). Atmospheric meteorological parameters were monitored with a Vaisala Automatic Weather Station (AWS) and the visibility was measured with a present weather sensor (Vaisala, model FD12P). More details on these permanent weather and

climate monitoring at Pallas Sammaltunturi site are given by Lohila et al. (2015).

### 2.5   Air mass back-trajectories

Air mass backward trajectories for the arrival height of 500 m a.g.l. were calculated with the NOAA HYSPLIT_4 model using the meteorological model data of the global CDC1 reanalysis data set (Draxler and Hess, 1997, 1998; Stein et al., 2015). Trajectories were calculated every six hours starting at 0:00, 6:00, 12:00 and 18:00 on UTC time (local winter time is UTC+2)

and by following the air mass 72h backwards in 1h time steps.



## 3   Results

### 3.1   Campaign overview

Arctic air masses prevailed during the campaign and correspondingly very low aerosol concentrations were measured (Figure 2). The 1h-averaged scattering coefficient measured with Aurora 4000 ($\sigma_{\mathrm{AUR4,635nm}}$) was $3.9 \pm 5.2$ Mm$^{-1}$, absorption coef-
ficient measured with MAAP ($\sigma_{\mathrm{MAAP,637nm}}$) was $0.09 \pm 0.08$ Mm$^{-1}$ and total particle number (PN) was $1254 \pm 1129$ cm$^{-3}$ (Table 2). Aerosol was highly scattering with a $\omega_{0,635nm} = 0.97$ and dominated by relatively large particles as suggested by the average $\alpha_{SP,450-635nm} = 1.2$.

The campaign was divided in two periods: Period 1 (June 19 – July 7) and Period 2 (July 7 – July 17), based on the measured meteorological and aerosol characteristics (Figure 2). During Period 1 the air masses originated in north-west and coastal
north-east Arctic (Figure 3). Precipitation was frequently observed and the visibility was low. The Pallas station, due to its high altitude, was then occasionally measuring inside a cloud. Those "in-cloud" data in Period 1, characterized by $\sigma_{\mathrm{AUR4,635nm}} <$ 1 Mm$^{-1}$ and a simultaneous decrease in visibility, were omitted from further analysis (see Figure 2). On the afternoon of July 7, a sudden increase in aerosol scattering accompanied by a persistent change in the meteorological situation was observed. During Period 2 the air masses originated in north-east Arctic (Figure 3) and the weather was partially cloudy with no further
precipitation observed at the measurement site. The daily average temperature slightly increased as compared to Period 1.

A notable increase in all measured aerosol extensive properties was seen in Period 2. The aerosol number concentration increased on average by 25% as compared to Period 1 (Table 2). The increase was the most pronounced in aerosol optical properties: extinction, scattering and absorption roughly doubled in Period 2 as compared to Period 1. The aerosol optical size related parameter $\alpha_{SP,450-635nm}$ increased an average from 1.3 to 1.5, suggesting a decreasing particle size. This simultaneous
increase in number concentration and decreasing particle optical size suggests that the secondary aerosol formation processes were recently activated in Period 2. Another aerosol intensive property, $\omega_{0,635nm}$, remained stable at around 0.97 throughout the campaign. A value of 0.97 represents the higher end of the $\omega_{0,\lambda}$ range that is typically observed at Pallas, and shows that strongly scattering, and only weakly absorbing, aerosol was measured throughout the campaign. The persistent Arctic air masses measured during the campaign is a plausible explanation for the high $\omega_{0,\lambda}$ values.

### 3.2   Absorbing aerosol characteristics

The average aerosol absorption coefficient $\sigma_{AP,630nm}$ during the EMPIR BC campaign ranged from 0.063 to 0.096 Mm$^{-1}$ in Period 1 (in cloud-free conditions) and from 0.116 to 0.151 Mm$^{-1}$ in Period 2 (Table 3), depending on the measurement method. The 25th to 75th percentiles extend a range of $0.04 - 0.13$ Mm$^{-1}$ (Period 1) and $0.06 - 0.18$ Mm$^{-1}$ (Period 2) (Figure 4). These values are in the lower end of what is expected at Pallas. Lihavainen et al. (2015) long-term analysis showed that
a $\sigma_{AP,\lambda}$ in a range of 0.1–1 Mm$^{-1}$ is typically measured during summer. However, the measured $\sigma_{AP,\lambda}$ values during the campaign are well representative of the summer Arctic absorbing aerosol in general (Schmeisser et al., 2018).

The climate impact of absorbing aerosol depends essentially on the aerosol mixing state in the atmosphere. At Pallas this was estimated by measuring the individual rBC particle mass and their coating using SP2. Winter season measurements on





absorbing aerosol mixing state at Pallas have revealed significant dependence on aerosol source, presenting a dominance of
the internally mixed mode (Raatikainen et al., 2015). However, in summer the absorbing aerosol sources in Pallas are very
different from those in winter and could significantly affect aerosol mixing state (Lihavainen et al., 2015; Raatikainen et al.,
2015). Figure 5 shows the measured lag-time between the peak of the scattering signal and the incandescence signal of rBC
containing particles. The lag-time is proportional to the evaporation time of the coating and is a qualitative measure of the
coating thickness (Metcalf et al., 2012). The results revealed that throughout the campaign two types of rBC particles were
present: thinly coated (Mode1) and more thickly coated (Mode2) (Figure 5). In Period 2 the fraction of more thickly coated
(Mode2) rBC particles increased, which is connected with an increasing aerosol atmospheric age. This suggests that the long-
range transported aerosol from the north-east of the Arctic in Period 2 contained an elevated fraction of absorbing aerosols
(Figure 3). In Period 1 the vast majority (70–100 %) of the rBC particles had a thin coating, suggesting that their origin was
rather local and not influenced by long-range transport and cloud-processing. This could be interpreted both as a sign of very
small rBC source contributions in the north-west region of the Arctic as well as an impact of effective particle removal by wet
scavenging during the transport.

### 3.3   Sensitivity of the methods at low concentrations

Results from the different filter-based methods agreed qualitatively well with each others but in absolute scale the differences
were significant. The highest measured average $\sigma_{AP,630nm}$ during Period 1 was 52% above the lowest measured average
$\sigma_{AP,630nm}$, and still in Period 2 a difference of 30% was observed in despite of the increased overall absorption level (Table 3).
The lowest $\sigma_{AP,\lambda}$ values were measured with COSMOS technique which also had the lowest variance of the five instruments
studied (Figure 4). This is expected since the sample pre-treatment in COSMOS heated inlet effectively removes the particle
light scattering coating thereby eliminating the related artifacts. Heating also decreases fluctuations by possible sample RH
variations. However, the sample modification also changes the aerosol interaction with light which then no longer corresponds
to its dry atmospheric state. Therefore, this method is primarily used to determine the mass of the absorbing refractory particles,
rather than the aerosol light absorption in the atmosphere. The highest concentrations were measured with AE33 technique.
While this is also the newest of the methods used, post-processing of the data and correction factors have been little studied and
thus subject to major uncertainties. However, the relatively small variance observed in AE33 data encourages the application
of this method at low concentrations.
330       In addition to the filter-based techniques, we estimated the $\sigma_{AP,\lambda}$ indirectly using the extinction minus scattering (EMS)
technique. The calculated $\sigma_{AP,\lambda}$ values with EMS$_1$, EMS$_2$ and EMS$_3$ method for Period 1 and Period 2 are presented in Table
3. They show large discrepancies between each other and the filter-based methods. The EMS methods overestimated the aerosol
light absorption up to 10-fold compared with the filter-based techniques, and their sensitivity was considered insufficient to
detect such low absorbing aerosol levels.





Sensitivity of different methods was estimated with signal-to-noise ratio (SNR) method, assuming that the detection limit of signal follows:

$$SNR = \frac{\mu}{\sigma} < 3, \tag{5}$$

where $\mu$ is the signal and $\sigma$ is its variance. Noise of the measured absorption was determined during a 6h period of clean, particle-free measurements. Data were analysed at different averaging times and the result is presented in Figure 6. The filter-
based methods are about 20-fold more sensitive than EMS and can separate the signal from noise at $\sigma_{AP,\lambda}$ values of around 0.05 Mm$^{-1}$ or more (Figure 6b). Instead, the EMS method is sensitive to a $\sigma_{AP,\lambda}$ only at around 1 Mm$^{-1}$ values or higher when using 1h-averaged data (Figure 6a). Out of the three EMS techniques applied, EMS$_3$ here shows the best SNR. However, EMS techniques which rely on a separate instrument for scattering and absorption are more widely characterized and documented in the literature due to their longer history in use (Strawa et al., 2003; Virkkula et al., 2005; Moosmuller et al., 2009). AE33 was the
least noisy filter-based method when short averaging times were used, whereas MAAP and COSMOS could be theoretically utilized to detect the smallest signal levels when data are averaged over several hours. In general, the different filter-based techniques showed a very similar $1/\sqrt{t}$ decay of SNR with an increasing averaging time (t).

### 3.4   Instrument comparison: linear relation

Linear correlation between the absorption coefficients measured with different techniques was studied using the Williamson-
York bivariate fitting method provided by Cantrell (2008). This method is less sensitive to outliers than the standard least-squares method, and considers that uncertainties can exist in both fitting variables which here is the case. The correlation statistics ($R^2$, slope and intercept) for the filter-based techniques are presented in Table 4. All correlations were statistically significant. Good correlation ($R^2 = 0.84$–$0.85$) of $\sigma_{AP,\lambda}$ from MAAP with AE33, PSAP and COSMOS was obtained. $\sigma_{MAAP,\lambda}$ correlation was nearly linear with $\sigma_{PSAP,\lambda}$, while a slope of around 0.7 was obtained with $\sigma_{AE33,\lambda}$ and $\sigma_{COSMOS,\lambda}$. Visual-
ization of these correlations (Fig. 7b-d) shows an apparent overestimation of AE33 at very low $\sigma_{AP,\lambda}$ values and further an underestimation of COSMOS with an increasing concentration, both in comparison to MAAP. $\sigma_{AE33,\lambda}$ and $\sigma_{COSMOS,\lambda}$ present the best mutual correlation (slope = 0.94 and $R^2 = 0.90$) which is marginally higher than correlations with $\sigma_{MAAP,\lambda}$. High scatter in $\sigma_{AE31,\lambda}$ is evident in Figure 7a, and coincides with the low the $R^2$ of 0.46 even when the correlation with $\sigma_{MAAP,\lambda}$ is fairly linear (slope = 1.13).
Given that the measured absorption coefficients are close to the detection limits of the instruments the correlations are reasonably good. Slopes and intercepts could be modified with a different selection of data outliers which at such low concentrations can be a sensitive choice. Correlations of $\sigma_{AP,\lambda}$ from filter-based with EMS methods were not calculated, except for the two examples shown in Figure 7e-f for MAAP correlation with EMS$_1$ and EMS$_3$. Significant overestimation of the slope and low correlation coefficient is evident in both examples, and explained by the higher detection limit of the EMS methods,
as previously shown.





## 3.5 Particle size and coating impact on measured absorption

Aerosol type differed between Period 1 and Period 2. The aerosol in Period 2 was 1) more numerous, 2) had smaller optical size, and 3) an increased rBC coating thickness. These three changes occured nearly simultaneously in our data. The increasing rBC particle coating thickness is argued to induce a lensing effect, which could intensify the absorption signal in filter-based instruments other than COSMOS (Fuller et al., 1999). In some previous studies the observed increase in aerosol absorption with atmospheric aging has been explained by the lensing effect of the added coating material (Schmid et al., 2006; Slowik et al., 2007; Kanaya et al., 2008). However, the actual magnitude of this effect is still uncertain (Cappa et al., 2012; Liu et al., 2017). The impact of the rBC aerosol mixing state on the springtime Arctic aerosol radiative forcing was recently estimated by Zanatta et al. (2018), suggesting a notable contribution from the lensing effect based on Svalbard Arctic data.

The Pallas data shows an increasing $\sigma_{\mathrm{MAAP},\lambda}$ in comparison to $\sigma_{\mathrm{COSMOS},\lambda}$ as a function of the particle coating thickness (Figure 8a). This could be interpreted as an indication of an increase in the lensing effect by the non-absorbing shell surrounding the absorbing core. However, when $\sigma_{\mathrm{COSMOS},\lambda}$ was compared with $\sigma_{\mathrm{PSAP},\lambda}$ an opposite result was obtained: more coating resulted in less absorption by PSAP (Figure 8b). This suggests a small over-compensation of the scattering artifact by the PSAP or under-compensation by the MAAP. AE31 and AE33 data in comparison with COSMOS did not show a clear tendency as a function of the coating thickness. Therefore, our data does not provide an unambiguous evidence of the importance of the lensing effect.

Variance in data decreased systematically with an increasing coating thickness. This simply follows from the simultaneous increase in total aerosol absorption which thus improves the signal-to-noise. The impact of the changing aerosol size distribution on the filter penetration depth artifact remains difficult to quantify, not only based on our data, but also in general. Figure 7 shows in color scale the concurrent change in aerosol optical size with the aerosol absorption. In other words, at low $\sigma_{AP,\lambda}$ values the particle optical size is larger and vice verse. Qualitatively, at least two size-dependent correlations are observed: PSAP $\sigma_{AP,\lambda}$ increases with respect to MAAP when particles size decreases and the opposite is observed in COSMOS versus MAAP. This adds another degree of freedom in our analysis.

The size effect, inevitably mixing with the lensing effect, was studied by comparing the non-volatilized sample (larger particles) measured by MAAP, AE33, PSAP and AE31 with the volatilized sample (smaller particles) measured by COSMOS. A systematic increase of this ratio with a decreasing aerosol optical size was observed using MAAP, AE31 and AE33 (Figure 9). PSAP showed no clear correlation but this could be related with the slight over-compensation shown in Figure 8b. The interpretation for the observed $\sigma_{AP,\lambda}$ dependency on aerosol optical size can be explained either with an increasing organic carbon content with $\alpha_{SP,\lambda}$ or with the difference in size of the measured volatilized and non-volatilized particles (Kondo et al., 2009). Either way, it is clear that the various filter artifacts might partly compensate for each other in some circumstances, and their impact on the results finally depends on the aerosol size distribution shape, composition and mixing, in accordance with several above cited publications.





## 4    Conclusions

Absorbing aerosol characteristics were measured using several techniques in EMPIR BC campaign at Pallas in northern Fin-
land. Measured aerosol concentrations were very low throughout the campaign. Air masses from north-west and coastal north-
east Arctic with persistent low pressure systems along with precipitation prevailed during the first 20 days (Period 1). A shift
to the direction from north-east Arctic was accompanied by a dry weather during the last 10 days of the campaign (Period 2).
Because of the differences in aerosol origin and transport, data for these two periods were analysed separately. In Period 2
the average aerosol light absorption coefficient roughly doubled as compared to Period 1, increasing from 0.07 to about 0.14
Mm$^{-1}$. It should, however, be noted that these levels are still very low compared to levels measured outside the Arctic. The
air mass change was also accompanied by a decrease in aerosol size in Period 2 suggesting the presence of secondary aerosol
particle formation mechanisms. The Arctic aerosol measured throughout the campaign was highly scattering with an average
single scattering albedo of 0.97.

The absorbing aerosol particles have a refractory Black Carbon core (rBC) which at Pallas was mixed in two modes: rBC
particles with thin coating connected with the local emissions and rBC particles with a thicker coating connected with the
long-range transport. The fraction of the latter ones increased during the last 10 days. This was attributed to a change in aerosol
source region along with the dry prevailing weather which led to a potential decrease in wet scavenging in aerosol transport in
Period 2. The changes in absorbing aerosol properties occurring around day 20 of the campaign were measured using several
instruments simultaneously.

Overall, the filter-based absorption instruments are shown to be a robust and sensitive method to measure absorption at low
concentrations such as in the Arctic. In the absorption coefficient range of this study: 0.04 to 0.18 Mm$^{-1}$ (representing the 25th
to 75th percentiles), the different instruments agreed approximately within 20%. PSAP showed the best linear correlation with
MAAP ((R$^2$ = 0.85), followed by AE33 and COSMOS (R2 = 0.84). The noisy data from AE31 resulted in a slightly lower, yet a
significant, correlation (R$^2$ = 0.46). COSMOS data had the lowest variation, followed by AE33 and PSAP. The signal-to-noise
analysis suggested that the 1h-averaging of data used here provides a sufficient sensitivity at $\sigma_{AP,\lambda} >$0.05 Mm$^{-1}$. In contrast,
the sensitivity of the indirect extinction minus scattering (EMS) technique was found insufficient at $\sigma_{AP,\lambda} <$1.0 Mm$^{-1}$. The
EMS methods when assessed against the MAAP absorption values overestimated aerosol absorption up to 10-fold.

The impact of the changing aerosol type around the measurement day 20 for the instruments response was not uniform.
An increase of $\sigma_{AP,\lambda}$ with a thicker coating on rBC aerosol was only observed in MAAP data. The ratio of non-volatilized
(measured with MAAP, AE33, AE31, PSAP) to volatilized (measured with COSMOS) $\sigma_{AP,\lambda}$ increased as a function of the
aerosol optical size ($\alpha_{AP,\lambda}$), which was interpreted as an indication of the aerosol size or chemistry related artifacts. However,
the simultaneously occurring multiple changes in aerosol characteristics: decrease in aerosol size and increase concentration
and coating, hinders a quantitative determination of the filter artifacts especially at such low concentrations. Our results address
the importance of long-term studies in varying field environments to unveil the site and aerosol type specific filter artifacts.
This study was a first of this kind made at Pallas.



In general, the results highlight the importance of good measurement practices and careful data post-processing when aiming at on qualitative knowledge on the absorbing aerosol concentrations in the Arctic or similar clean environments. Future studies should focus on providing the means for field instruments reference and calibration methods to improve the accuracy of the filter-based methods.

*Data availability.* Pallas WMO Global Atmospheric Watch (GAW) station, part of the Aerosols, Clouds, and Trace gases Research InfraStructure (ACTRIS), submits aerosol number, size, scattering and absorption measurement data annually to EBAS database operated at the Norwegian Institute for Air Research (NILU) (http://ebas.nilu.no). These data are available at no cost and can be used in agreement with the ACTRIS Data Policy statement. The specific data measured during EMPIR BC campaign are available for scientific use upon request. Aurora 4000 data are available at TROPOS, CAPSssa data at DEMOKRITOS and all other data at FMI by contacting the corresponding data
owner and co-author in this manuscript.

*Author contributions.* EA, HS, JB, MG, KE, TM and AH participated in planning the campaign. EA, JB, HS and MG made the measurements. EA analysed the filter-based instrument absorption data and all auxiliary data with contributions from AV and JB. JB analysed SP2 data. MG and KE analysed CAPS data. TM analysed Aurora nephelometer data. SO, YK and HS analysed COSMOS data. EA prepared the paper with contributions from all authors.

*Competing interests.* Authors do not declare any competing interests.

*Acknowledgements.* We greatly acknowledge the funding from the 16ENV02 Black Carbon project of the European Union through the European Metrology Programme for Innovation and Research (EMPIR). The financial support of the ACTRIS by the European Union's Horizon 2020 research and innovation programme under grant agreement no. 654109 is also gratefully acknowledged. This research was also supported by Academy of Finland via project NABCEA (grant no. 29664) and by Business Finland via project BC Footprint (grant no.
49402-201040). This work was also supported by the Environmental Research and Technology Development Fund (2-2003), and the Arctic Challenge for Sustainability II (ArCS II) project of Japan.



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

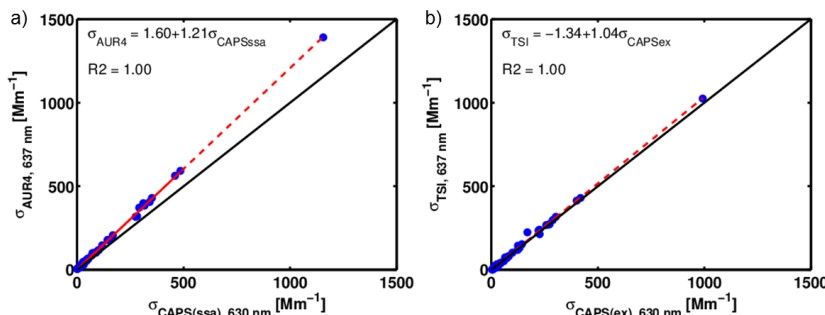

**Figure 1.** Results from the three ammonium sulphate calibrations that were performed during the campaign. Data are averaged to 5-min. In y-axis the aerosol scattering measured by a) the Aurora 4000 nephelometer at $\lambda$ 635 nm and b) the TSI nephelometer interpolated to a $\lambda$ 635 nm and in x-axis the extinction measured by a) the CAPSssa at $\lambda$ 630 nm and b) the CAPSex $\lambda$ 630 nm. The slopes from the calibrations were used to determine the loss-correction term for both CAPS instruments in comparison to the two nephelometers.

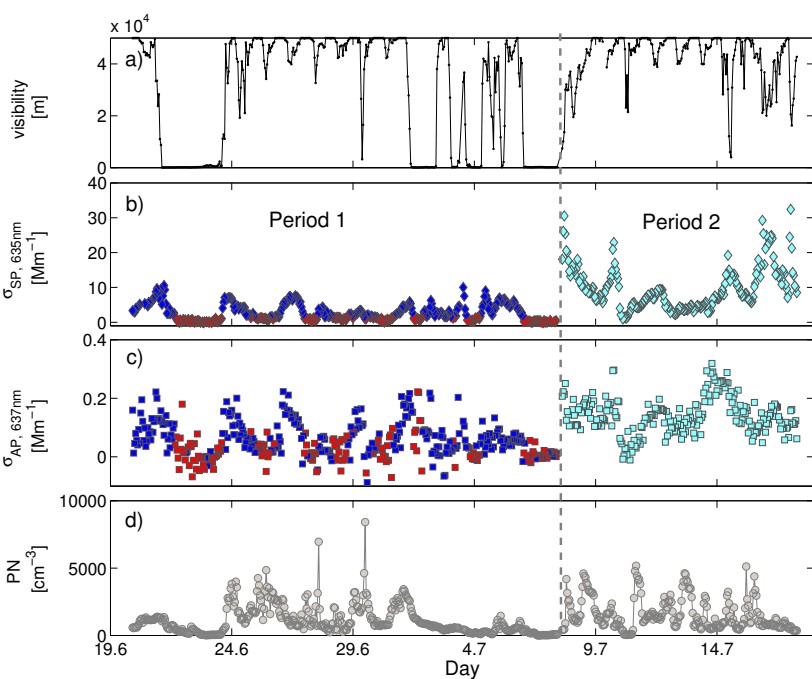

**Figure 2.** Hourly averaged a) visibility, b) scattering coefficient at $\lambda = 635$ nm, d) absorption coefficient at $\lambda = 637$ nm and d) total aerosol particle number concentration (PN) during the campaign. The campaign was divided in two parts: Period 1 (dark-blue points in panels b) and c)) and Period 2 (light-blue points in panels b) and c)). Decreased visibility coincided with rainy days during which the aerosol scattering was significantly reduced. These data (marked with red points in panels b) and c)) were discarded from further analysis due to cloud screening.

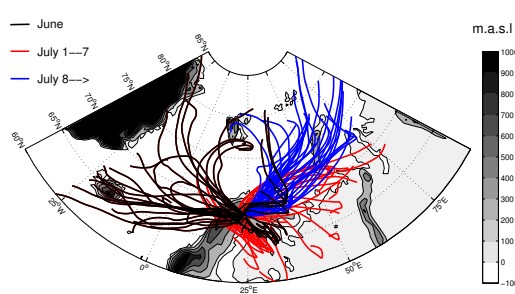

**Figure 3.** Air mass back-trajectories at arrival height of 500 m a.g.l. for the campaign period calculated every 6hs, 72h backwards. Back-trajectories in June, in July 1–7, and during the Period 2 (after July 7) are plotted with different colors: black, red and blue, respectively.



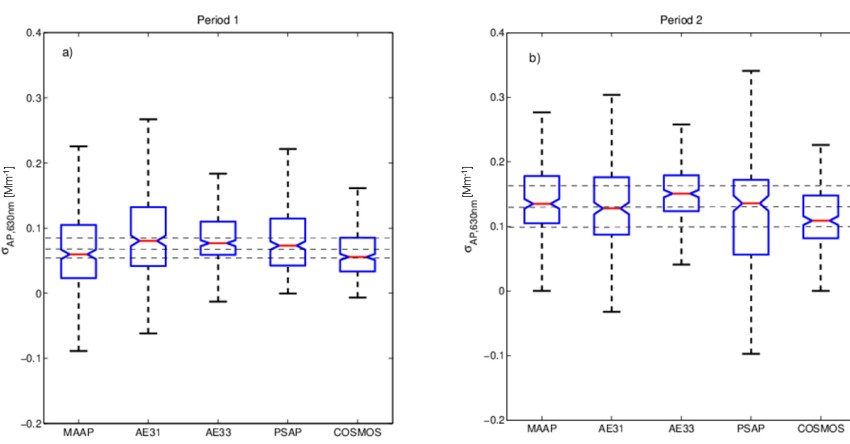

**Figure 4.** The vertical black lines show the range of 1h-averaged absorption coefficient values in a) Period 1 and b) Period 2 with the five different filter-based instruments named in x-label. The red lines show the medians and the blue boxes the 25th and 75th percentiles. The horizontal dotted black lines present the overall median in Period 1 and Period 2 $\pm$ 25%.



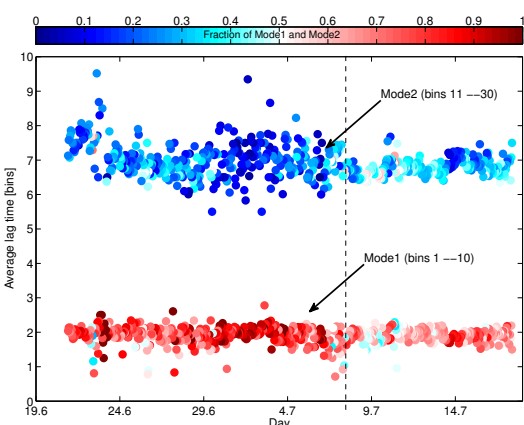

**Figure 5.** The average SP2 lag-time [bins] of the two modes observed in data: Mode1 (average of bins 1–10) and Mode2 (average of bins 11–30). The color indicates the fraction of the modes such that blue = fraction < 0.5; red = fraction > 0.5.

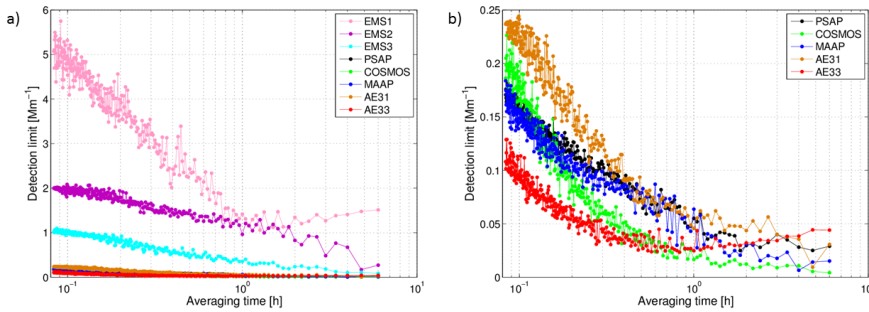

**Figure 6.** The detection limit of absorption coefficient (y-axis), calculated as 3 times the signal variance $\sigma$, for the eight instruments and methods (represented with colors) as a function of the data averaging time (x-axis).





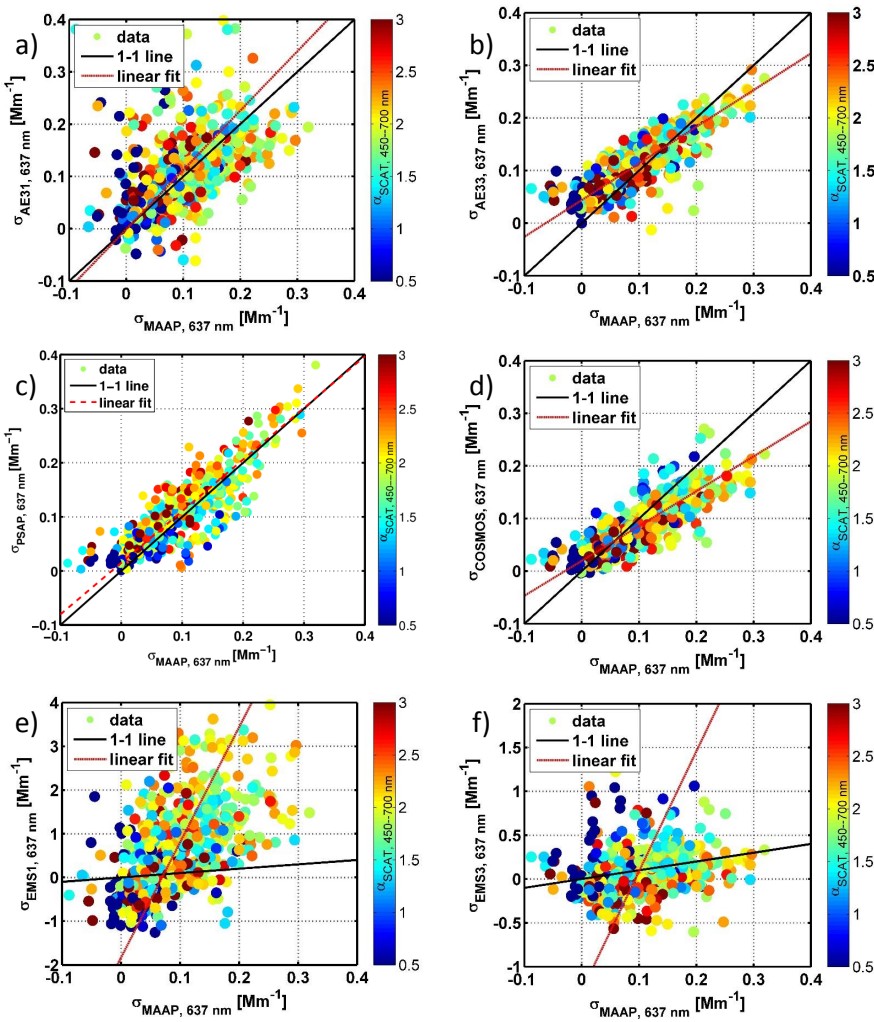

**Figure 7.** Correlation between the 1h-averaged absorption coefficient measured with MAAP and a) AE31, b) AE33, c) PSAP, d) COSMOS, e) EMS1 method, and f) EMS3 method calculated at 637 nm. Corresponding correlation coefficients and $R^2$-statistics are presented in Table 4.



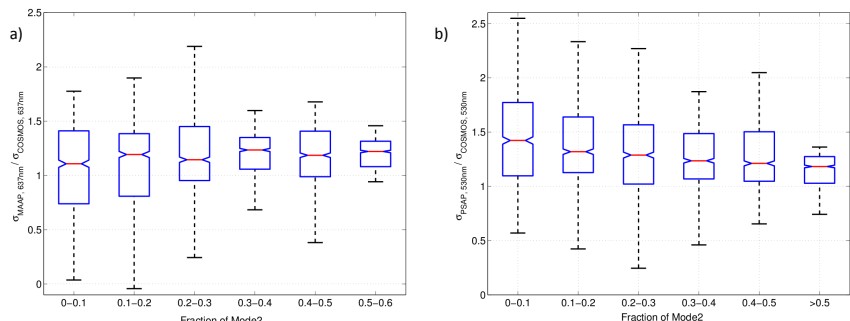

**Figure 8.** The ratio of the absorption coefficient measured with a) MAAP and COSMOS at $\lambda$=637nm and b) PSAP and COSMOS at $\lambda$=530nm as a function of the Mode2 fraction measured by SP2. Increasing fraction of Mode2 represents an increase in rBC particle coating thickness.





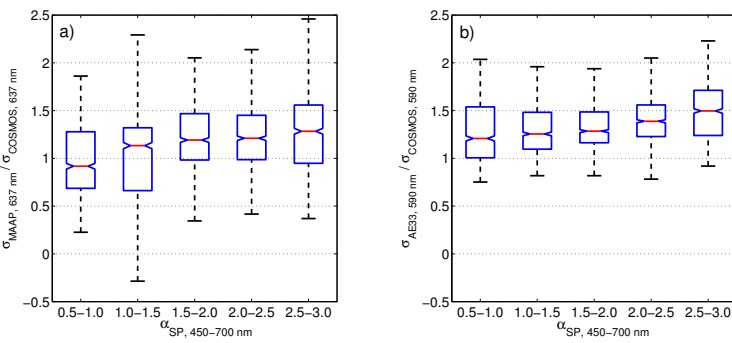

**Figure 9.** The ratio of the absorption coefficient measured with a) MAAP and COSMOS at $\lambda$=637nm and b) AE33 and COSMOS at $\lambda$=590nm as a function of the Ångström exponent of scattering ($\alpha_{SP}$). An increasing $\alpha_{SP}$ represents a decrease in aerosol optical size.



**Table 1.** The campaign instrumentation presented in columns: (1) instrument appreviation, (2) variable measured (scattering = sca; extinction = ext; refractory Black Carbon = rBC; equivalent Black Carbon = eBC; particle number concentration = PN), (3) measurement time resolution, (4) flow rate, (5) instrument inlet (cut-size 10 $\mu$m = PM10 or Total inlet = TOT).

| Instrument | Variable | Time res [s] | Volumetric Flow [LPM] | Inlet |
|---|---|---|---|---|
| AE31 | eBC | 300 | 4.5 | TOT |
| AE33 | eBC | 60 | 5.8 | PM10 |
| MAAP | eBC | 60 | 8.0 | PM10 |
| PSAP | eBC | 1 | 1.0 | PM10 |
| COSMOS | rBC | 60 | 0.7 | PM10 |
| SP2 | rBC | 1 | 0.1 | TOT |
| CAPSex | ext | 5 | 0.8 | PM10 |
| CAPSssa | ext | 1 | 0.9 | PM10 |
| NEPH AUR4 | sca | 10 | 5.8 (via AE33) | PM10 |
| NEPH TSI3 | sca | 300 | 8.0 (via MAAP) | PM10 |
| CPC 3772 | PN | 1 | 1.02 | PM10 |



**Table 2.** Average ($\pm$ standard deviation) aerosol characteristics during the measurement Period 1 and the Period 2: Particle number (PN), Extinction coefficient ($\sigma_{EP,630nm}$, CAPSssa), Scattering coefficient ($\sigma_{SP,635nm}$, AUR4), Absorption coefficient ($\sigma_{AP,637nm}$, MAAP), Scattering Ångström exponent ($\alpha_{SP,450-635nm}$, AUR4) and Single scattering albedo ($\omega_{0,635nm}$). Period1 is further divided into two parts: the whole period (all) and the cloud free part of it (no-cloud). In further analysis, only the cloud free part of Period 1 is included.

| Variable | All data | Period 1 (all) | Period 1 (no-cloud) | Period 2 |
|---|---|---|---|---|
| PN [cm$^{-3}$] | $1254 \pm 1129$ | $1081 \pm 1059$ | $1251 \pm 1096$ | $1566 \pm 1182$ |
| $\sigma_{EP,630nm}$ [Mm$^{-1}$] | $4.5 \pm 5.3$ | $2.6 \pm 4.5$ | $3.6 \pm 5.0$ | $8.0 \pm 5.2$ |
| $\sigma_{SP,635nm}$ [Mm$^{-1}$] | $3.9 \pm 5.2$ | $2.5 \pm 5.3$ | $3.2 \pm 6.1$ | $6.3 \pm 4.0$ |
| $\sigma_{AP,637nm}$ x100 [Mm$^{-1}$] | $8.5 \pm 7.5$ | $5.5 \pm 6.2$ | $6.8 \pm 6.4$ | $14.1 \pm 6.5$ |
| $\alpha_{SP,450-635nm}$ | $1.2 \pm 0.8$ | $1.1 \pm 1.0$ | $1.3 \pm 0.8$ | $1.5 \pm 0.5$ |
| $\omega_{0,635nm}$ | $0.97 \pm 0.09$ | $0.97 \pm 0.11$ | $0.98 \pm 0.02$ | $0.97 \pm 0.02$ |





**Table 3.** Average ± standard deviation of the absorption coefficient measured with different instruments in Period 1 and Period 2.

| Instrument | Period 1 $\sigma_{AP,630nm}$ [Mm]$^{-1}$x100 | Period 2 $\sigma_{AP,630nm}$ [Mm]$^{-1}$x100 |
|---|---|---|
| MAAP | 6.8 ± 6.4 | 14.1 ± 6.5 |
| AE31 | 9.6 ± 7.8 | 13.6 ± 8.1 |
| AE33 | 8.8 ± 5.0 | 15.1 ± 4.7 |
| PSAP | 8.5 ± 5.7 | 12.2 ± 9.4 |
| COSMOS | 6.3 ± 4.2 | 11.6 ± 4.8 |
| EMS$_1$ | 30.6 ± 85.3 | 130.8 ± 91.9 |
| EMS$_2$ | 14.5 ± 47.9 | 71.4 ± 94.3 |
| EMS$_3$ | 0.5 ± 40.8 | 17.2 ± 26.6 |





**Table 4.** Linear correlation coefficients and $R^2$-values between 1h-average absorption values from the five filter-based techniques. All presented correlations are statistically highly significant ($p < 0.001$).

| Method | MAAP | AE31 | AE33 | PSAP | COSMOS |
|---|---|---|---|---|---|
| MAAP (637nm) | | 1.13x-0.01 | 0.70x+0.04 | 0.95x+0.02 | 0.66x+0.02 |
| ($R^2$x100) | | (46.3) | (84.6) | (85.4) | (84.1) |
| AE31 (660nm) | 0.84x+0.01 | | 0.47x+0.06 | 0.86x+0.01 | 0.45x+0.03 |
| ($R^2$x100) | (46.3) | | (47.6) | (45.4) | (48.1) |
| AE33 (660nm) | 1.40x-0.06 | 2.13x-0.12 | | 1.51x-0.07 | 0.94x-0.02 |
| ($R^2$x100) | (84.6) | (47.6) | | (68.6) | (89.7) |
| PSAP (660nm) | 1.04x-0.01 | 1.16x-0.01 | 0.66x+0.04 | | 0.60x+0.02 |
| ($R^2$x100) | (85.4) | (45.4) | (68.6) | | (65.8) |
| COSMOS (565nm) | 1.48x-0.03 | 1.81x-0.05 | 1.06x+0.03 | 1.63x-0.04 | |
| ($R^2$x100) | (84.1) | (48.1) | (89.7) | (65.9) | |



## Appendix A: Symbols and abbreviations

**Table A1.** Summary of symbols and abbreviations frequently used in the manuscript text.

| Symbol | Explanation |
|---|---|
| $\lambda$ | Wavelength of light. |
| $\sigma_{SP,\lambda}$ | Scattering coefficient at wavelength $\lambda$. |
| $\sigma_{AP,\lambda}$ | Absorption coefficient at wavelength $\lambda$. |
| $\sigma_{EP,\lambda}$ | Extinction coefficient at wavelength $\lambda$. |
| $\sigma_{0,\lambda}$ | Absorption coefficient at wavelength $\lambda$ as directly reported by instrument. |
| $\sigma_{INST,\lambda}$ | Absorption coefficient at wavelength $\lambda$ after corrections measured with INST. |
| INST | Abbreviation of the absorption measurement instrument (see section 2 Methodology for a complete list of instruments used). |
| $\omega_{0,\lambda}$ | Single-scattering albedo at wavelength $\lambda$. |
| $\alpha_{SP,\lambda}$ | Ångström exponent of scattering at wavelength $\lambda$. |
| $\alpha_{AP,\lambda}$ | Ångström exponent of absorption at wavelength $\lambda$. |
| $EMS_n$ | Extinction minus scattering technique to measure absorption, where n = 1, 2 or 3, |
| | referring to a pair of instruments used (see section 2 Methodology for details). |