# Peer review of "Absorption instruments inter-comparison campaign at the Arctic Pallas station"

_Atmospheric Measurement Techniques, 2020_

## Referee Comment (RC1) · Anonymous Referee #1 · 24 Nov 2020

REVIEW OF CHARACTERIZING THE ARCTIC ABSORBING AEROSOL WITH MULTI-INSTRUMENT OBSERVATIONS

OVERVIEW The work of Asmi et al. presents optical properties of Arctic aerosol measured with a wide array of instruments. Considering the global intensive use of the considered instruments, understanding and quantifying the issues of each instrument in order to optimize its performances is an essential task. However, the present manuscript lays between a technical assessment of the performances of filter-based absorption photometers and a survey of arctic aerosol optical properties. Thus, the objectives of the manuscript are not very clear nor are the scientific and the technical conclusions. The dataset is of undoubtable value, the authors have, nonetheless, clarify their technical or scientific message. I do not recommend the publication of the manuscript in its

present form. However, with the hope that my comments will be helpful to the authors, I suggest a major rethinking of the manuscript.

MAJOR COMMENTS My biggest concern is represented by the overall "take home message", which is hard to grasp. The "Campaign overview" and "Absorbing aerosol characteristic" are not of scientific relevance, since similar results have been widely presented in previous and more comprehensive works. Hence, the characterization of aerosol properties and airmass origin, which does not have a clear impact on the instrumental comparison, adds only confusion. As an example, the distinction between period 1 and period 2 is not used in the more technical part. The 5 filter-based absorption photometers agree to a variable degree. The actual causes are, however, not clear or not investigated. As a matter of fact, Section 3.5 provides, citing the manuscript, "unambiguous evidence" on the impact of mixing and size on optical measurements.

The SP2 is used to provide the degree of internal mixing of rBC particles. Why this is done with the lag time technique and not with the LEO fit. Although both are prone to large uncertainty, the first is only qualitative. If my understanding is right, the Mode1 Mode2 classes are based on a lag time distribution. It appears that the lag time analysis was applied to all BC particles. This might cause substantial biass in the fraction of thickly and thinly coated particles. By limiting the analysis to BC cores falling in the detection range of scattering detector, the fraction of thickly coated particles should degrees (see specific comments below). Hence, the very interesting and also surprising results shown in 3.5 might be wrong.

SPECIFIC COMMENTS

L: line of text; F: figure; S: section

Title: from the title the reader might expect a soot optical characterization on a large Arctic scale. I suggest to slightly modify the tittle specifying the location of measurements.

L32: Worth citing the NILU report : https://www.amap.no/work-area/document/3058

L49-51: I find the statement about MAC and eBC a bit out of place and might generate confusion. eBC should be rather mentioned in the filter-based instrument paragraph. Since the eBC and MAC are mostly used for the filter-based instruments, I suggest to mention them a bit earlier.

L78: It would be appropriate to shortly summarize the goal of the manuscript in this last paragraph, or clearly state that this work was performed within the framework of EMPIR BC, if this is the case.

L95-100: description of goals and objectives does not belong to method sections, more to introduction.

L100-109: in this subchapter there are many abbreviations of the various instruments, which might become overwhelming and confusing to a non-expert reader. I suggest to move this plumbing description in a separate subchapter after the instrumental description.

L145: I would not use the abbreviation MAC to describe the coefficient used internally by the AE31.

L110-126: only the SSA is really described. The MAC is briefly described elsewhere in the text. I suggest to rework a bit this part in order to provide a more systematic and inclusive description of all optical properties.

L159-160: This is actually a very good point

L192: Continuous soot monitoring system . . . capital or non capital?

L193-195: Many periods in this sentence, writing could be smoother and more enjoyable. This is a constant feature of the paper. I suggest the authors to work a bit on it.

L206: not sure if capital letters for Black Carbon are needed.

L210: The work of Lim focussed on SP2 measurement in snow. I think there are better references: (Laborde et al., 2012).

L211-213: as stated: "This technique is very sensitive but does not measure particle light absorption as such, and therefore, a direct comparison with other absorption measurement techniques is not straightforward.". So, what the SP2 is used for in this work?

L215: provide CAPS full name

L220-221: Despite very recent and in review, I suggest giving a look to (Modini et al., 2020).

L262-270: these two subsections (2.4.10 and 2.5) are very short. I suggest combining them together with the plumbing description (L100-110) into a unique subsection: "Additional tools and methods"

L273-277: these numbers are not very useful without any reference for comparison. Are these pristine, background, polluted conditions for Finnish Arctic? Considering the influence of different airmasses and, thus, different aerosol loads and properties, averaged values are definitely not of interest. I suggest removing this paragraph.

L278-279: the distinction of the two periods in not very clear. Especially considering the backtrajectories shown in F3 (see related comment).

L288-291: "The aerosol optical size related parameter", is confusing. Simply use the symbol or "Angstrom exponent". I would be careful to jump into conclusions: the absence of precipitation and thus wet scavenging (both from nucleation and impaction) might cause increase of number concentration and diameter decrease. Moreover optical diameter measurements are not available.

L292: typically observed...add reference and potentially a value.

L296-298:What do you mean with "average $\sigma$AP630nm"? Average between all instrument? Unclear. Same at L319

L307-309: lag-time description. . .move this to technical section. Is the lag applied to all rBC signal or a to a specific rBC diameter range? Although this measurement does appear to be only qualitative here and does not need supreme robustness in this case, applying a lower limit to rBC particles diameter (let's say above the detection limit of the scattering signal) will reduce the number of thickly coated BC cores (the weak incandescence signals with no coating (total particle size below 150-200 nm) will not be seen by scattering detector). Not compulsory, but worth trying. This might change your statement at L310.

L321-322: from my understanding COSMOS directly provide a "eBC" with the cinstant COSMOS-MAC value. Here absorption coefficient is presented, which MAC was used. Worth specify in the respective technical section.

L320-334: These paragraphs so not provide relevant information; Or, at least, it is hard to understand what the authors want to show.

L337: in the equation there is a ">". Is this correct or it should be " * "? L361-365: Why ? Is the low sensitivity the sole explanation to the bad correlation between EMS and MAAP ? This is quite interesting since CAPS and nephelometer should not suffer from filter matrix-effect.

L379-380: What do you mean with "a clear tendency". I suggest plotting the AE31 and AE33 results in figure 8. Same for figure 9.

L393-395: the increase of absorbing organic carbon could be seen with tha absorption angstrom exponent.

S3.1-3.2: Since the main focus is absorption I suggest merging these two sections

S3.4: is the analysis done on the full campaign or on a selected period?

F1: this figure is partially needed to understand the scientific message of the paper. I

would move it in the supplementary.

F3: I find the legend and caption a bit confusing. These are the 2 considered periods: Period 1 (June 19 – July 7) and Period 2 (July 7 – July 17). The legend does not reflect this partitioning

F5: what the meaning of bins is?

F6: Define the difference between panels

F7 the colour scale does not provide useful additional information. Scattering coefficient is not even mentioned in the text. I wonder if SSA might provide a more info.

F8 the x-axis label is a bit confusing, I suggest to use a more understandable label "Fraction of thickly coated rBC"

REFERENCE Laborde, M., Schnaiter, M., Linke, C., Saathoff, H., Naumann, K.-H., Möhler, O., Berlenz, S., Wagner, U., Taylor, J. W., Liu, D., Flynn, M., Allan, J. D., Coe, H., Heimerl, K., Dahlkötter, F., Weinzierl, B., Wollny, A. G., Zanatta, M., Cozic, J., Laj, P., Hitzenberger, R., Schwarz, J. P. and Gysel, M.: Single Particle Soot Photometer intercomparison at the AIDA chamber, Atmos Meas Tech, 5(12), 3077–3097, doi:10.5194/amt-5-3077-2012, 2012. Modini, R. L., Corbin, J. C., Brem, B. T., Irwin, M., Bertò, M., Pileci, R. E., Fetfatzis, P., Eleftheriadis, K., Henzing, B., Moerman, M. M., Liu, F., Müller, T. and Gysel-Beer, M.: Detailed characterization of the CAPS single scattering albedo monitor (CAPS PMssa) as a field-deployable instrument for measuring aerosol light absorption with the extinction-minus-scattering method, Atmospheric Meas. Tech. Discuss., 1–56, doi:https://doi.org/10.5194/amt-2020-292, 2020.

---

## Referee Comment (RC2) · Anonymous Referee #2 · 15 Jan 2021

This manuscript describes measurements of aerosol concentrations and optical properties (absorption, scattering, extinction) measured during a month-long period at the Pallas ground site in northern Finland. Filter-based aerosol absorption instruments include two models of aethelometers, a MAAP, a PSAP, and COSMOS. Extinction and scattering coefficients are obtained from CAPS and nephelometer instruments, respectively. Information on black carbon mass and coating thickness is obtained from an SP2, while particle number concentrations are measured by a CPC. In total, this is a comprehensive aerosol measurement suite!

The primary focus of the study is to examine the instrument consistency during two time periods – Period 1 is characterized by relatively low aerosol scattering coefficients, while Period 2 sees higher particle scattering coefficients. Aerosol absorption

coefficients and number concentrations are similar across both periods, after removing cloud/fog artifacts when ambient visibility was greatly reduced. Backtrajectories are included to provide context for air mass history, which indicate consistently northeast-erly winds during Period 2, while the wind directions during Period 1 are much more variable. Despite some degree of variability between the two time periods, the overall aerosol absorption, scattering, and number concentrations are quite low, which would be expected for the Pallas region (far from continental pollution aerosol sources). This challenges many of the instrument comparisons because the measurements are close to the the lower limits of detection, and it appears that the data are highly-averaged to reduce noise (which is appropriate and fine).

Overall, the manuscript is well written, enjoyable to read, and nicely describes the Pallas scientific instruments and the observations obtained during this time period; although, the scientific importance of these observations is not clear from the paper. The depth of analysis is pretty shallow as only a short period of time is being examined and the mean concentrations shown in Table 2 tend to be close to zero (with large standard deviations relative to the magnitude of the mean), which makes it hard to draw conclusions regarding instrument agreement/disagreement. I'm also unsure if AMT is the appropriate journal for this manuscript. All instruments are commercially available and have previously been described in the literature, and there don't seem to be novel conclusions related to their performance and/or operation that would be informative to the broader scientific community. Since I do not think that the present paper would be acceptable for publication without bringing in additional data and completely rewriting/reframing its scope, I recommend that it be rejected for publication in AMT.

**Specific Comments**:

1) As stated above, the depth of analysis in this paper is low. Simply comparing the data from multiple, filter-based aerosol optical property measurements for a month and reporting summary statistics that largely overlap with zero provides little value regarding the instrument operation or the remote measurement site characteristics. One way to increase the depth of analysis might be to reframe the paper to describe the annual aerosol climatology relevant to the Pallas site. This might include monthly and seasonal data from satellites and/or models that provide some long-term context for the site measurements. It would also be great if more in situ data could be included beyond a single month; although, I recognize that that might be prohibitive. Having more in situ data might allow for more dynamic range in the aerosol abundance and optical properties. Another approach to increase the depth of analysis might be to examine the Pallas aerosol instrument suite response to laboratory-generated aerosol under very controlled conditions to understand their interconsistency and accuracy. These results could then be compared to the ambient measurements to understand/evaluate the instrument performance. These are just a couple of ideas for the authors' consideration in a potential future manuscript.

2) The sensitivities and lower detection limits given in the abstract (and elsewhere) need to include the averaging interval. I'm assuming that 0.05 $Mm^{-1}$ was achieved by averaging for > 1 hr. What are the overall instrument accuracies?

3) On Line 24, what is meant by the statement, "additional activation of secondary particle formation mechanisms"?

4) Strike first sentence on Line 29 as not relevant. The second sentence is referencing a filter-based measurement method, which I'm assuming is of aerosol absorption yes? Also please correct typos on Line 29 (begun -> began) and Line 31 (graphic -> graphitic).

5) On Line 96, it is noted that the Arctic aerosol absorption measurements are "demanding". Is this related to the low concentrations due to a lack of local combustion sources? If so, it might be worthwhile to clarify that, while also noting that such, low-aerosol conditions are also not unique to the Arctic.

6) Is the mass absorption coefficient (MAC) defined on Line 145 the same as the MAC used on Line 200? On Line 146 it is called a "wavelength dependent specific attenuation".

7) What value(s) of the MAC were applied as mentioned on Line 201?

8) On Lines 248 (and elsewhere), I find the extinction-minus-scattering (EMS 1  2) terminology confusing because this is neither an instrument nor a complex method. Rather, it's just a difference between two values. I suggest that it would be worth denoting as $\sigma_{CAPSex} - \sigma_{TSI}$ and $\sigma_{CAPSssa} - \sigma_{AUR4}$ to help avoid confusion and to be consistent with the notation in Figure 1.

9) On Line 257, it is noted that differences in CAPS instrument response may be due in part to "discrepancies in inlet tubing sizes and flow rates", which seems unlikely to me. Can this sentence be clarified to explain how these differences would affect the aerosol measurements?

10) On Lines 274-275, Table 2, and elsewhere, I note that the magnitudes of arithmetic standard deviations often exceed the magnitudes of the arithmetic means, which gives implies non-physical, negative values. Is this because of instrument noise that results in a negative baseline or is this because the observational data are not normally distributed about the mean. At least in the case of the CPC data reported in Table 2, it would seem that the latter explanation is correct, in which case it would be appropriate to report the summary statistics as either a geometric mean */ one geometric standard deviation or as median and percentiles.

11) Rather than using the qualitative SP2 lag times, can the data be reanalyzed to provide quantitative coating thicknesses? I think that this is important and would help

to increase the paper depth of analysis.

12) What support is there for the statement made on Lines 310-11 that the more thickly coated aerosols have longer atmospheric ages?

13) The final conclusions on Lines 429-434 are not really connected to the data that are presented in this manuscript. What site- and aerosol-specific filter artifacts were presented and overcome? How were good measurement practices and careful data post-processing demonstrated in this study that differ from conventional techniques? To put it bluntly, what is new or novel from the present paper in terms of measurement techniques?

14) There seem to be too many significant figures presented in Table 3, which imply the absorption coefficient measurements can be made with a precision of 0.001 Mm$^{-1}$. I realize that the're a lot of averaging going on here to help tamp down the noise, but I suspect that at least the last digit is probably not significant.

---

## Author Comment (AC1) · 30 Apr 2021

We wish to thank reviewer for their excellent comments and suggestions that have greatly helped us to re-structure and improve the presentation of our campaign results in the form of this manuscript. We hope that the reviewers find the changes acceptable and the current version of the manuscript more appropriate for publication in AMT. We have modified the manuscript content significantly. We summarize below the major changes that were done and answer the detailed reviewers' comments below the summary.

Major changes, summary:

The modified manuscript focuses entirely on instruments capabilities for detecting low concentrations in a pristine field environment. All discussion on atmospheric relevance and implications of the measured aerosol concentrations, including the analysis of air mass origin and Arctic BC mixing state were removed. All figures, tables and instrument correlations were re-calculated using 100% of campaign data (i.e. not separately for periods 1, 2 or neglecting some periods). This affected slightly the correlation statistics described in subsection 3.4 (current 3.3). Results section structure was modified such that the former subsection 3.3 (on detection limits) was moved at the beginning of section 3 (current 3.1) and Allan variances were calculated and used to determine the lowest averaging times of the instruments. Subsection 3.1 is now followed by modified and compressed former sections 3.1 and 3.2 (campaign overview and observed absorption values). The former section 3.5 (Particle size and coating impact on measured absorption) was removed completely and a new short section 3.4 with a simplistic MAC-value calculation was added. The main conclusions of the modified manuscript are: - filter-based methods are sensitive to detect absorption coefficients down to around 0.01 Mm-1 level (1-sigma) while the use of EMS method requires around 10-fold higher absorption coefficient values - Arctic summer absorption values were most of the time between 0.06-0.1 Mm-1 in our study, which is well above the lowest detection limits of filter-based instruments, but too low for EMS methods - Even at these low concentrations, the absorption values measured by different filter-based instruments show a good linear correlation, confirming both the accuracy and the precision seem to be adequate. Here the exception was the absorption measured by COSMOS instrument, where the pre-treatment of the sample can, as expected, modify the measured absorption. This led to a slope clearly below 1 between MAAP and COSMOS absorption. - Mass-absorption cross section of 16 m2 g-1 was determined for MAAP, when compared to a residual BC mass measured by SP2. This is well in-line with the MAC values obtained in a comprehensive study in the Arctic by Ohata et al., 2020.

Following references were added: Werle et al., 1993; Ohata et al., 2020; Allan et al.,

1966; Hagler et al., 2011; Springston and Sedlacek, 2007; Laing et al., 2020; Modini et al., 2021; Torseth et al., 2019; Bond et al., 2006; Jacobson 2001; Sinha et al., 2017.

Our detailed answers to Referee #1:

The work of Asmi et al. presents optical properties of Arctic aerosol measured with a wide array of instruments. Considering the global intensive use of the considered instruments, understanding and quantifying the issues of each instrument in order to optimize its performances is an essential task. However, the present manuscript lays between a technical assessment of the performances of filter-based absorption photometers and a survey of arctic aerosol optical properties. Thus, the objectives of the manuscript are not very clear nor are the scientific and the technical conclusions. The dataset is of undoubtable value, the authors have, nonetheless, clarify their technical or scientific message. I do not recommend the publication of the manuscript in its present form. However, with the hope that my comments will be helpful to the authors, I suggest a major rethinking of the manuscript.

ANSWER: We thank reviewer for their time and effort dedicated towards our work. We largely share the reviewers' concerns and appreciate the very useful comments and suggestions made to improve the content and the structure of the manuscript. We have done our best to modify the text accordingly and hope that the reviewer finds the current form of the manuscript more appropriate for publication in the journal.

MAJOR COMMENTS My biggest concern is represented by the overall "take home message", which is hard to grasp. The "Campaign overview" and "Absorbing aerosol characteristic" are not of scientific relevance, since similar results have been widely presented in previous and more comprehensive works. Hence, the characterization of aerosol properties and airmass origin, which does not have a clear impact on the instrumental comparison, adds only confusion. As an example, the distinction between period 1 and period 2 is not used in the more technical part. The 5 filter-based absorption photometers agree to a variable degree. The actual causes are, however, not clear

or not investigated. As a matter of fact, Section 3.5 provides, citing the manuscript, "unambiguous evidence" on the impact of mixing and size on optical measurements. The SP2 is used to provide the degree of internal mixing of rBC particles. Why this is done with the lag time technique and not with the LEO fit. Although both are prone to large uncertainty, the first is only qualitative. If my understanding is right, the Mode1 Mode2 classes are based on a lag time distribution. It appears that the lag time analysis was applied to all BC particles. This might cause substantial bias in the fraction of thickly and thinly coated particles. By limiting the analysis to BC cores falling in the detection range of scattering detector, the fraction of thickly coated particles should degrees (see specific comments below). Hence, the very interesting and also surprising results shown in 3.5 might be wrong.

ANSWER: We do share reviewers concerns regarding the "take home message" and content of the manuscript. To respond to these concerns, we gave a thorough thinking on the ms structure and decided to completely re-structure the manuscript and sharpen the objectives. In the modified manuscript the objectives are to study the 1) absorption instrument's stability and detection limits with respect to the atmospheric concentrations commonly measured in pristine environments and 2) instruments accuracy when operated at the edge of their lower detection limits. In addition, we calculate 3) an Arctic-specific mass absorption cross-section (MAC) value based on SP2 as an rBC reference, and MAAP as an absorption reference. Any previous publication provides such a comprehensive instruments comparison in such a pristine field environment, yet, the very same instruments are often applied in these environments and data is compared (Scheisser et al., 2018; Tørseth et al., 2019; Ohata et al., 2020). To keep the "take home message" simple and sharp, we have removed all discussion on the climatic relevance of the results, and removed the sections on air mass analysis, aerosol mixing state and optical properties of atmospheric aerosol. We do agree with the reviewer that those have been more thoroughly characterized in several previous publications.

SPECIFIC COMMENTS: line of text; F: figure; S: section Title: from the title the reader might expect a soot optical characterization on a large Arctic scale. I suggest to slightly modify the tittle specifying the location of measurements.

ANSWER: We accommodated the title with the new content and focus of the ms, taking into account this suggestion. The new title is "Absorption instruments inter-comparison campaign at the Arctic Pallas station"

L32: Worth citing the NILU report : https://www.amap.no/work-area/document/3058

ANSWER: Cited.

L49-51: I find the statement about MAC and eBC a bit out of place and might generate confusion. eBC should be rather mentioned in the filter-based instrument paragraph. Since the eBC and MAC are mostly used for the filter-based instruments, I suggest to mention them a bit earlier.

ANSWER: This sentence was moved in the end of the previous paragraph.

L78: It would be appropriate to shortly summarize the goal of the manuscript in this last paragraph, or clearly state that this work was performed within the framework of EMPIR BC, if this is the case.

ANSWER: The following text was added: "The goal was to test the stability, accuracy and detection capabilities of the commonly available absorption measurement methods focusing on the filter-based techniques, and to conclude on their applicability to pristine environments. To our knowledge, this is the most comprehensive absorption and BC mass measurement instrument parallel field comparison done in the Arctic."

L95-100: description of goals and objectives does not belong to method sections, more to introduction.

ANSWER: Done.

L100-109: in this subchapter there are many abbreviations of the various instruments,

which might become overwhelming and confusing to a non-expert reader. I suggest to move this plumbing description in a separate subchapter after the instrumental description.

ANSWER: These were moved to a new subsection "Sampling"

L145: I would not use the abbreviation MAC to describe the coefficient used internally by the AE31.

ANSWER: Agree. Term "MAC" was removed.

L110-126: only the SSA is really described. The MAC is briefly described elsewhere in the text. I suggest to rework a bit this part in order to provide a more systematic and inclusive description of all optical properties.

ANSWER: The text was re-checked and explanations on absorption coefficient and MAC were added.

L159-160: This is actually a very good point

ANSWER: I would assume to have some recommendation on this soon. Also the information given here adds to the knowledge.

L192: Continuous soot monitoring system...capital or non capital?

ANSWER: Non-capital, thanks for pointing this out.

L193-195: Many periods in this sentence, writing could be smoother and more enjoyable. This is a constant feature of the paper. I suggest the authors to work a bit on it.

ANSWER: We hope to have improved the writing, here and in other sections too.

L206: not sure if capital letters for Black Carbon are needed.

ANSWER: True. This, and other similar mistakes were corrected.

L210: The work of Lim focussed on SP2 measurement in snow. I think there are better references: (Laborde et al., 2012).

ANSWER: Reference changed.

L211-213: as stated: "This technique is very sensitive but does not measure particle light absorption as such, and therefore, a direct comparison with other absorption measurement techniques is not straightforward.". So, what the SP2 is used for in this work?

ANSWER: The original idea to use SP2 was to describe the atmospheric mixing state of the absorbing aerosol which affects the absorption enhancement. In the modified manuscript version, SP2 is used as a rBC mass reference to define the value of MAC. This is now clearly said in ms text.

L215: provide CAPS full name ANSWER: Done.

L220-221: Despite very recent and in review, I suggest giving a look to (Modini et al.,2020).

ANSWER: Thank you, this was very interesting. The work by Modini et al. is now cited at the discussion of the CAPS accuracy, stability and error sources.

L262-270: these two subsections (2.4.10 and 2.5) are very short. I suggest combining them together with the plumbing description (L100-110) into a unique subsection: "Additional tools and methods"

ANSWER: In the modified ms some of these sections (air mass analysis, CPC) were completely removed. The remaining notes on AWS were connected with the suggested new section entitled "Sampling and environment".

L273-277: these numbers are not very useful without any reference for comparison. Are these pristine, background, polluted conditions for Finnish Arctic? Considering the influence of different airmasses and, thus, different aerosol loads and properties,

averaged values are definitely not of interest. I suggest removing this paragraph.

ANSWER: In the modified ms the absorption time series and the averaged values are compared to the instruments detection limits, and further to those typically measured in the Arctic. We feel that it is important to make the point that around the Arctic, such low concentrations do exist, justifying the need to understand the instruments capabilities to measure such low concentrations. In the end of section 3.2 we state: "The measured absorption coefficient values are in the lower end of that typically observed at Pallas site. Lihavainen et al., 2015 long-term analysis showed that a sigma_ap in Pallas summer ranges between 0.1-1 Mm-1, where the lower values represent the clean Arctic air flow. Thus, the measured sigma_ap values during the campaign are well representative of the values measured around the Arctic during summer (Schmeisser et al., 2018)."

L278-279: the distinction of the two periods in not very clear. Especially considering the back trajectories shown in F3 (see related comment).

ANSWER: The two periods are no longer separated and data is no longer filtered.

L288-291: "The aerosol optical size related parameter", is confusing. Simply use the symbol or "Angstrom exponent". I would be careful to jump into conclusions: the absence of precipitation and thus wet scavenging (both from nucleation and impaction) might cause increase of number concentration and diameter decrease. Moreover optical diameter measurements are not available.

ANSWER: True. All this discussion was removed.

L292: typically observed...add reference and potentially a value.

ANSWER: This was removed. Our main reference to Pallas aerosol optical properties in ms is Lihavainen et al., 2015.

L296-298: What do you mean with "average$\sigma$AP630nm"? Average between all instrument? Unclear. Same at L319

ANSWER: Average sigma_AP is always an average value measured with one instrument. We tried to formulate this more clear and make appropriate references to Tables and Figures with numbers.

L307-309: lag-time description...move this to technical section. Is the lag applied to all rBC signal or a to a specific rBC diameter range? Although this measurement does appear to be only qualitative here and does not need supreme robustness in this case, applying a lower limit to rBC particles diameter (let's say above the detection limit of the scattering signal) will reduce the number of thickly coated BC cores (the weak incandescence signals with no coating (total particle size below 150-200 nm) will not be seen by scattering detector). Not compulsory, but worth trying. This might change your statement at L310.

ANSWER: The lag-time analysis was removed completely.

L321-322: from my understanding COSMOS directly provide a "eBC" with the constant COSMOS-MAC value. Here absorption coefficient is presented, which MAC was used. Worth specify in the respective technical section.

ANSWER: Done.

L320-334: These paragraphs so not provide relevant information; Or, at least, it is hard to understand what the authors want to show.

ANSWER: We agree that this was repetitive with previous section. The point was to provide information on the atmospheric sigma_ap values measured by different instruments and their deviation. This is now completely re-structured and combined with previous sections.

L337: in the equation there is a ">". Is this correct or it should be " * "?

ANSWER: This equation is removed. The symbol was correct, though.

L361-365: Why? Is the low sensitivity the sole explanation to the bad correlation between EMS and MAAP ? This is quite interesting since CAPS and nephelometer should not suffer from filter matrix-effect.

ANSWER: This is a good question, and we are happy that you find it as an interesting observation to point out. It is one of the main conclusions of the manuscript. Indeed, these instrumetns should not suffer from matrix-effects but rather the components noise and drift and truncation errors can explain the relatively high lowest detection limit of the techniques. Important is, that at high SSA values such as here (0.97) the absorption is defined from a subtraction between two big number which can amplify the errors (Modini et al., 2021).

L379-380: What do you mean with "a clear tendency". I suggest plotting the AE31 and AE33 results in figure 8. Same for figure 9. L393-395: the increase of absorbing organic carbon could be seen with the absorption angstrom exponent.

ANSWER: The last section was removed.

S3.1-3.2: Since the main focus is absorption I suggest merging these two sections ANSWER: Agree, and done.

S3.4: is the analysis done on the full campaign or on a selected period? ANSWER: Full campaign.

F1: this figure is partially needed to understand the scientific message of the paper. IC5would move it in the supplementary. ANSWER: Done.

F3: I find the legend and caption a bit confusing. These are the 2 considered periods: Period 1 (June 19 – July 7) and Period 2 (July 7 – July 17). The legend does not reflect this partitioning F5: what the meaning of bins is? F6: Define the difference between panels F7 the colour scale does not provide useful additional information. Scattering coefficient is not even mentioned in the text. I wonder if SSA might provide a more info. F8 the x-axis label is a bit confusing, I suggest to use a more understandable label"Fraction of thickly coated rBC"

ANSWER: Most figures were either removed or completely modified.

REFERENCE Laborde, M., Schnaiter, M., Linke, C., Saathoff, H., Naumann, K.-H.,Möhler, O., Berlenz, S., Wagner, U., Taylor, J. W., Liu, D., Flynn, M., Allan, J. D.,Coe, H., Heimerl, K., Dahlkötter, F., Weinzierl, B., Wollny, A. G., Zanatta, M., Co-zic,J., Laj, P., Hitzenberger, R., Schwarz, J. P. and Gysel, M.: Single Particle Soot Pho-tometer intercomparison at the AIDA chamber, Atmos Meas Tech, 5(12), 3077–3097,doi:10.5194/amt-5-3077-2012, 2012. Modini, R. L., Corbin, J. C., Brem, B. T., Irwin,M., Bertò, M., Pileci, R. E., Fetfatzis, P., Eleftheriadis, K., Henzing, B., Moerman, M.M., Liu, F., Müller, T. and Gysel-Beer, M.: Detailed characterization of the CAPS single scattering albedo monitor (CAPS PMssa) as a field-deployable instrument for measuring aerosol light absorption with the extinction-minus-scattering method, Atmo-spheric Meas. Tech. Discuss., 1–56, doi:https://doi.org/10.5194/amt-2020-292, 2020.

ANSWER: References added.

---

## Author Comment (AC2) · 30 Apr 2021

We wish to thank the reviewer for their excellent comments and suggestions that have greatly helped us to re-structure and improve the presentation of our campaign results in the form of this manuscript. We hope that the reviewers find the changes acceptable and the current version of the manuscript more appropriate for publication in AMT. We have modified the manuscript content significantly. We summarize below the major changes that were done and answer the detailed reviewers' comments below the summary.

Major changes, summary:

The modified manuscript focuses entirely on instruments capabilities for detecting low concentrations in a pristine field environment. All discussion on atmospheric relevance and implications of the measured aerosol concentrations, including the analysis of air mass origin and Arctic BC mixing state were removed. All figures, tables and instrument correlations were re-calculated using 100% of campaign data (i.e. not separately for periods 1, 2 or neglecting some periods). This affected slightly the correlation statistics described in subsection 3.4 (current 3.3). Results section structure was modified such that the former subsection 3.3 (on detection limits) was moved at the beginning of section 3 (current 3.1) and Allan variances were calculated and used to determine the lowest averaging times of the instruments. Subsection 3.1 is now followed by modified and compressed former sections 3.1 and 3.2 (campaign overview and observed absorption values). The former section 3.5 (Particle size and coating impact on measured absorption) was removed completely and a new short section 3.4 with a simplistic MAC-value calculation was added. The main conclusions of the modified manuscript are: - filter-based methods are sensitive to detect absorption coefficients down to around 0.01 Mm-1 level (1-sigma) while the use of EMS method requires around 10-fold higher absorption coefficient values - Arctic summer absorption values were most of the time between 0.06-0.1 Mm-1 in our study, which is well above the lowest detection limits of filter-based instruments, but too low for EMS methods - Even at these low concentrations, the absorption values measured by different filter-based instruments show a good linear correlation, confirming both the accuracy and the precision seem to be adequate. Here the exception was the absorption measured by COSMOS instrument, where the pre-treatment of the sample can, as expected, modify the measured absorption. This led to a slope clearly below 1 between MAAP and COSMOS absorption. - Mass-absorption cross section of 16 m2 g-1 was determined for MAAP, when compared to a residual BC mass measured by SP2. This is well in-line with the MAC values obtained in a comprehensive study in the Arctic by Ohata et al., 2020.

Following references were added: Werle et al., 1993; Ohata et al., 2020; Allan et al.,

1966; Hagler et al., 2011; Springston and Sedlacek, 2007; Laing et al., 2020; Modini et al., 2021; Torseth et al., 2019; Bond et al., 2006; Jacobson 2001; Sinha et al., 2017.

Our detailed answers to Referee #2:

This manuscript describes measurements of aerosol concentrations and optical properties (absorption, scattering, extinction) measured during a month-long period at the Pallas ground site in northern Finland. Filter-based aerosol absorption instruments include two models of aethelometers, a MAAP, a PSAP, and COSMOS. Extinction and scattering coefficients are obtained from CAPS and nephelometer instruments, respectively. Information on black carbon mass and coating thickness is obtained from an SP2, while particle number concentrations are measured by a CPC. In total, this is a comprehensive aerosol measurement suite! The primary focus of the study is to examine the instrument consistency during two time periods – Period 1 is characterized by relatively low aerosol scattering coefficients, while Period 2 sees higher particle scattering coefficients. Aerosol absorption coefficients and number concentrations are similar across both periods, after removing cloud/fog artifacts when ambient visibility was greatly reduced. Backtrajectories are included to provide context for air mass history, which indicate consistently northeasterly winds during Period 2, while the wind directions during Period 1 are much more variable. Despite some degree of variability between the two time periods, the overall aerosol absorption, scattering, and number concentrations are quite low, which would be expected for the Pallas region (far from continental pollution aerosol sources). This challenges many of the instrument comparisons because the measurements are close to the lower limits of detection, and it appears that the data are highly averaged to reduce noise (which is appropriate and fine). Overall, the manuscript is well written, enjoyable to read, and nicely describes the Pallas scientific instruments and the observations obtained during this time period; although, the scientific importance of these observations is not clear from the paper. The depth of analysis is pretty shallow as only a short period of time is being examined and the mean concentrations shown in Table 2 tend to be close to zero (with

large standard deviations relative to the magnitude of the mean), which makes it hard to draw conclusions regarding instrument agreement/disagreement. I'm also unsure if AMT is the appropriate journal for this manuscript. All instruments are commercially available and have previously been described in the literature, and there don't seem to be novel conclusions related to their performance and/or operation that would be informative to the broader scientific community. Since I do not think that the present paper would be acceptable for publication without bringing in additional data and completely rewriting/reframing its scope, I recommend that it be rejected for publication in AMT.

ANSWER: We appreciate your kind words and an excellent summary on our work. We take the message that the manuscript in its previous form is rather confusing and presents no clear value for the community, for which we have completely re-written the results section. Many of your excellent suggestions have been considered in the modified manuscript, and for example the Periods 1 and 2 are no longer separated, and the focus is not on aerosol climatic relevance, but rather on instruments ability to measure in such pristine environments, for which this campaign data serves as a unique data set and can show a broader value. We hope that you find our changes adequate.

Specific Comments: 1) As stated above, the depth of analysis in this paper is low. Simply comparing the data from multiple, filter-based aerosol optical property measurements for a month and reporting summary statistics that largely overlap with zero provides little value regarding the instrument operation or the remote measurement site characteristics. One way to increase the depth of analysis might be to reframe the paper to describe the annual aerosol climatology relevant to the Pallas site. This might include monthly and seasonal data from satellites and/or models that provide some long-term context for the site measurements. It would also be great if more in situ data could be included beyond a single month; although, I recognize that that might be prohibitive. Having more in situ data might allow for more dynamic range in the aerosol abundance and optical properties. Another approach to increase the

depth of analysis might be to examine the Pallas aerosol instrument suite response to laboratory-generated aerosol under very controlled conditions to understand their interconsistency and accuracy. These results could then be compared to the ambient measurements to understand/evaluate the instrument performance. These are just a couple of ideas for the authors' consideration in a potential future manuscript.

ANSWER: Thank you for your excellent suggestion on how to modify the focus of the ms. Finally, we took the decision to present the campaign results in a slightly different context, but with adding no data. We hope that you find this acceptable. One reason is that there are several previous long-term studies made at Pallas and other Arctic sites, but very view studies have focused on the instruments inter-comparisons in pristine field conditions. Secondly, we aim at reporting a laboratory characterization on these very same instruments. However, the results do not fit in this paper and it would change the scope completely, giving very little meaning to present the atmospheric inter-comparison, which is here the main focus. There are previous laboratory studies, too.

2) The sensitivities and lower detection limits given in the abstract (and elsewhere) need to include the averaging interval. I'm assuming that 0.05 Mm-1 as achieved by averaging for > 1 hr. What are the overall instrument accuracies?

ANSWER: Yes, these were added. Defining the accuracy is a question of a reference. Here, we used MAAP as an "absorption reference" and SP2 as a "mass reference", however understanding that these can not provide real references as such. The PSAP provided the best comparison to MAAP.

3) On Line 24, what is meant by the statement, "additional activation of secondary particle formation mechanisms"?

ANSWER: Removed.

4) Strike first sentence on Line 29 as not relevant. The second sentence is referencing a filter-based measurement method, which I'm assuming is of aerosol absorption yes? Also please correct typos on Line 29 (begun -> began) and Line 31 (graphic ->graphitic).

ANSWER: These were corrected as suggested.

5) On Line 96, it is noted that the Arctic aerosol absorption measurements are "demanding". Is this related to the low concentrations due to a lack of local combustion sources? If so, it might be worthwhile to clarify that, while also noting that such, low-aerosol conditions are also not unique to the Arctic.

ANSWER: This part was re-formulated to refer to pristine environments where techniques suffer from signal-to-noise challenges.

6) Is the mass absorption coefficient (MAC) defined on Line 145 the same as the MAC used on Line 200? On Line 146 it is called a "wavelength dependent specific attenuation".

ANSWER: MAC was removed from line 145, because it is not exactly the same. Thank you for notice.

7) What value(s) of the MAC were applied as mentioned on Line 201?

ANSWER: COSMOS MAC was 8.73 m2 g-1. This was added.

8) On Lines 248 (and elsewhere), I find the extinction-minus-scattering (EMS 1 2) terminology confusing because this is neither an instrument nor a complex method. Rather, it's just a difference between two values. I suggest that it would be worth denoting as $\sigma CAPSex - \sigma TSI$ and $\sigma CAPSssa - \sigma AUR4$ to help avoid confusion and to be consistent with the notation in Figure 1.

ANSWER: We are aware that this name has been in use only for a short time, partly because the technique as such is still relatively little applied. However, some previous work (e.g. Modini et al., 2021) use the name "EMS" and we would therefore wish to

keep it as it is, to clarify that the method is the same as in their ms.

9) On Line 257, it is noted that differences in CAPS instrument response may be due in part to "discrepancies in inlet tubing sizes and flow rates", which seems unlikely to me. Can this sentence be clarified to explain how these differences would affect the aerosol measurements?

ANSWER: We agree that the disagreement between the sampling line lengths and tubes was a negligible source of error and is unable to explain much of the difference. As we mention that the nephelometer sampling settings were slightly different, this could have a minor impact. This was re-phased now more clearly in the text.

10) On Lines 274-275, Table 2, and elsewhere, I note that the magnitudes of arithmetic standard deviations often exceed the magnitudes of the arithmetic means, which gives implies non-physical, negative values. Is this because of instrument noise that results in a negative baseline or is this because the observational data are not normally distributed about the mean. At least in the case of the CPC data reported in Table 2, it would seem that the latter explanation is correct, in which case it would be appropriate to report the summary statistics as either a geometric mean */ one geometric standard deviation or as median and percentiles.

ANSWER: You are correct that in some cases data are clearly not normally distributed and the std values lead to unphysical interpretations. We have now preferred the use of median and quartiles values, rather than averages and std in most of the ms tables and figures.

11) Rather than using the qualitative SP2 lag times, can the data be reanalyzed to provide quantitative coating thicknesses? I think that this is important and would help to increase the paper depth of analysis.

ANSWER: In the modified ms version, the aerosol mixing state is no longer discussed since the interpretation of this data (section 3.5) was too uncertain and speculative with

such low concentrations and short campaign time. We hope that you understand this decision.

12) What support is there for the statement made on Lines 310-11 that the more thickly coated aerosols have longer atmospheric ages?

ANSWER: Referring to our previous answer, this section was removed from the modified ms.

13) The final conclusions on Lines 429-434 are not really connected to the data that are presented in this manuscript. What site- and aerosol-specific filter artifacts were presented and overcome? How were good measurement practices and careful data post-processing demonstrated in this study that differ from conventional techniques? To put it bluntly, what is new or novel from the present paper in terms of measurement techniques?

ANSWER: We removed this sentence and aimed at sharpening the conclusions according to the new ms structure, also to answer better "what is new and novel in this paper".

14) There seem to be too many significant figures presented in Table 3, which imply the absorption coefficient measurements can be made with a precision of 0.001 Mm-1. I realize that the're a lot of averaging going on here to help tamp down the noise, but I suspect that at least the last digit is probably not significant.

ANSWER: This is probably very much true. We know that these 1h-average values are measured with instruments with a of lower detection limit on the order of $\sim 0.01 - 0.1$ Mm-1. What is the accuracy, is currently unknown since we do not have a reference method for aerosol absorption.

---

## Editor Decision (ED1)

Dr Asmi:

Thank you for the changes to the manuscript in response to the second review. The manuscript is now ready for publication in AMT following technical edits. The only really substantive change I request is in the last three sentences of the conclusions, which are rather cryptic and not specific.

Thanks for your responsiveness to the referees. I think this manuscript has substantially improved as it has progressed through revisions.

Technical edits:

1) Line 1. Change to "during a one-month field campaign"
2) Line 3. Change to "instruments' " (possessive)
3) Line 9. Change to "data were" (data is a plural noun)
4) Line 12. Change to "in the COSMOS inlet".
5) Line 13. Change to "A scattering correction"
6) Line 15. Clearly state what you mean by "best correlations". Slope closest to 1? Highest correlation coefficient?
7) Line 16. Change to "The sample pre-treatment in the COSMOS instrument resulted in the lowest fitted slope."
8) Line 18. Change to "method was not adequate to measure the low absorption values found at the Pallas site."
9) Line 19. Change to "the lowest absorption at which the EMS signal could be distinguished from the noise"
10) Line 20. Change to "cross-section (MAC) value measured was calculated using the MAAP and a single particle soot photometer (SP2), resulting in a MAC value of 16.0 m2g-1 +/- x.x." Please add the standard deviation of the MAC measured.
11) Line 25. Change to "filter-based aerosol absorption".
12) Line 30. Replace "Meanwhile" with "However".
13) Line 31. Change to "filter tape mass loading and the interference by aerosol"
14) Line 34. Change to "of the light source".
15) Line 48. Change to "allows derivation of the aerosol".
16) Line 54. Change "convergence" to "agreement".
17) Line 56. Don't capitalize "Organic Carbon"--it's not a place or product name.
18) Line 58. Change to "absorbing particles are analyzed".
19) Line 62. Don't capitalize "Polar Regions".
20) Line 62. Change to "Aerosol light absorption, "
21) Line 65. Change to "yet instrument inter-comparisons are few."
22) Line 66. Change to "showed good agreement".
23) Line 67. Change to "did not include aethalometer or MAAP instruments. Co-located aethometers"
24) Line 69. Change to "filter-based instruments in pristine field environments are lacking, leading to a poorly quantified".

25) Line 73. Change to "used co-located measurements".
26) Line 74. Change to "but a parallel comparison of all relevant instruments at one site has never been performed."
27) Line 74. Change to "would help estimate the uncertainties associated with these measurements and improve understanding of reported differences in baseline absorption at different Arctic stations."
28) Line 76. Can you provide a reference for the EMPIR BC project?
29) Line 81. Replace "conclude on" with "evaluate".
30) Line 84. Remove "The".
31) Line 89. Remove "also".
32) Line 91. Remove "earlier".
33) Line 94. Change to "instruments' " (possessive).
34) Line 96. Change "target to achieve" to "measure".
35) Line 98. Remove the sentence, "Sigma_AP,lambda is a measure of the . . . ." I don't think this sentence says anything useful.
36) Line 99. Change "This cross-section" to "Sigma_AP,lambda" (in symbols).
37) Line 100. Change to "normalized by particle mass, yielding a simple factor called".
38) Line 106. Change to "the ratio of the aerosol scattering coefficient sigma_SP,lambda to the aerosol extinction coefficient sigma_EP,lambda"
39) Line 115-116. What is this "simple relation"? Please delete the part of the sentence about variation and high-noise, or explain it more clearly.
40) Line 119. Change to "aerosol extinction, and one instrument that measures refractory BC, are used."
41) Line 122. Change to "with a volumetric flow calibrator (Gilian model xxx, city, country).
42) Line 124. To which instruments was the Bond et al. correction applied?
43) Line 126. Change to "The aethalometer Model AE31 (Magee Scientific Inc., city, country).
44) Line 127. Are these broad-spectrum measurements centered at these wavelengths, or are they made with lasers? Please state the bandwidth of this and other instruments (e.g., PSAP, COSMOS).
45) Line 132. Change to "The AE31 change the filter spot".
46) Line 134. Change to "converted with a wavelength-dependent specific attenuation and MAC values".
47) Line 139. Change to "station-specific"
48) Line 161-162. Please report band-widths for the wavelengths.
49) Line 164. Is this volumetric l/min, or mass-based (STP)?
50) Line 165. Change to "The PSAP records".
51) Line 179. How uncertain? Can you give a rough estimate?
51) Line 179. Delete everything in the sentence following "Figure 7".
52) Line 186. Change to "Due to elimination of most aerosol scattering artifacts and lensing enhancements of absorption, this method is typically".
53) Line 206. What is "CAPSex"? You define a CAPS PMex, but not a CAPSex.
54) Line 212. Change to "calibration-free".
55) Line 215. Change to "requires initial calibration"

56) Line 216. Change to "The extinction signal from each instrument was calibrated using the nephelometer (Sect. 2.3.8) at the beginning".
57) Line 225. What is meant by the "(180)" and "(90)" values? Total scattering is integrated across all angles (0-180 degrees, minus truncation), and backscattering from a nephelometer is hemispheric backscattering integrated from 90 to 180 degrees (minus truncation).
58) Line 228. Change to "gas at the beginning and end of the campaign."
59) Line 245. Change to "calibrations yielded correction factors of 1.21".
60) Line 247. Change to "The two CAPS used identical"
61) Line 248. Delete "(reference points)".
62) Line 249. Change to "discrepancies between the nephelometers were very minor (~x%) in comparison with those between the CAPS instruments and the nephelometers (~y%)."
63) Line 252. Change "The bulk" to "most".
64) Line 252. Don't capitalize "particulate matter".
65) Line 255. Change to "cylindrically symmetric exit tubes".
66) Eq. 5. How is the "t" in the left hand side of the equation related to the "j" on the right hand side? Is t the time at point j? Please clarify.
67) Line 271. Change to "during a 6-hour period of clean".
68) Line 277. Change to "a white-noise-dominated system".
69) Line 279. Change to "systematic drifts".
70) Line 283-284. I don't understand what this sentence means or accomplishes. Maybe change to "It is known that even the single-instrument EMS approach is subject to considerable uncertainty at low values of absorption (Modini et al., 2021)."
71) Line 288. Change to "each other, the instruments' detection limits".
72) Line 289. Change "predetermined" to "stated".
73) Line 293. Change to "measure the atmospheric aerosol in the Pallas campaign."
74) Line 297. Change to "The values of sigma_AP,630nm measured by the EMS methods differed 10-fold, giving a range".
75) Line 298. Delete the sentence beginning "In qualitative terms".
76) Line 300. Maybe at the end of this sentence add, "noise numbers than also have significant bias and drift relative to one another."
77) Line 301. Change to "increased slightly during the second half".
78) Line 302. What is "sigma_EX,630 nm"? See comment #38 above.
79) Line 304. Change to "wavelength-dependent".
80) Line 209. Change to "the campaign ranged from 0.07 to 0.09".
81) Line 312. Change to "instruments' " (possessive).
82) Line 313. Change to "with the lowest inter-quartile range (Fig. 3, Table 2)."
83) Line 314. Change to "effectively removes light-scattering particles and coatings, thereby".
84) Line 325. Change to "The calculated standard deviation around the average sigma_AP,lambda for all three EMS methods encompassed zero (Table 3).
85) Line 328. Change to "The absorption coefficients measured during this project were in the lower end of those typically observed at the Pallas site. Long-term analysis (Lihavainen et al., 2015) showed that the".
86) Line 333. Replace "conclusions" with "analysis".
87) Line 334. Change to "during the Pallas campaign represent well those observed".

88) Line 338. Change "bright" with "white". "Bright" implies intensity (e.g., a bright red light), while "white" implies color.

89) Line 342. Delete "between each others".

90) Line 344. Change to "has been widely utilized in the past as a practical field reference method (provide literature reference)."

91) Line 353. Change to "done for the PSAP data, which adds noise from the nephelometer measurement (literature reference), is a plausible explanation".

92) Line 356. I was very confused here, because the $r^2$ values already stated were 0.87 and 0.85. Do you mean "slope of the fitted line" instead of "correlation"?

93) Line 362. The sentence beginning "In case of the PSAP" makes no sense to me. What is "this"? What discrepancies at high single scatter albedos? Please clarify and rewrite.

94) Line 364. Change "correlations" to "correlation coefficients ($R^2$)".

95) Line 365. Suggest you add a sentence: "This suggests that the filter techniques examined here can provide useful data with an accuracy of ~40% for 1-2 hour averages even at the very low absorption values encountered. However, slopes and intercepts are sensitive to different selection of data outliers at such low absorption values."

96) Line 366. Suggest inserting a paragraph break, then beginning the paragraph with, "Comparison of sigma_AP,lambda measured with."

97) Line 367. Suggest adding to this new paragraph, "Our observations do not support the use of EMS methods in such pristine environments."

98) Line 371. Replace "SP2, extending the range" with "SP2 over the range". You aren't extending anything.

99) Line 379. Change to "Pallas site using the MAAP and SP2".

100) Line 376. In addition to the Ohata et al. (2020) reference, Zanatta et al. (2018) may provide useful support for such high MAC values.

*Zanatta, M., Laj, P., Gysel, M., Baltensperger, U., Vratolis, S., Eleftheriadis, K., Kondo, Y., Dubuisson, P., Winiarek, V., Kazadzis, S., Tunved, P., and Jacobi, H.-W.: Effects of mixing state on optical and radiative properties of black carbon in the European Arctic, Atmos. Chem. Phys., 18, 14037–14057, https://doi.org/10.5194/acp-18-14037-2018, 2018.*

101) Line 387. Replace "were in range" with "ranged from".

102) Line 290. Change to "EMS methods are not usable at sigma_AP,lambda < 0.1 Mm^-1, at least for the high omega_0 values encountered in this study."

103) Line 393. Where did "15-20%" come from? Slopes for two of the techniques vs. MAAP were 0.62 and 0.68, which suggests ~40% accuracy.

104) Line 394. Change to "The values we measured were at the lower edge of absorption typically measured at the Pallas site, but well represent Arctic summer conditions encountered at other sites."

105) Line 397. Change to "values, but likely led to".

106) Line 397. Change to "noisy AE31 data".

107) Line 402. Change "correlation" to "fitted".

108) Line 404. Change to "and is consistent with previous values measured during Arctic studies (literature references)."

109) Line 407. Change to "Sufficient averaging of data to reach minimum instrument detection limits and a utilization of".

110) Line 409. Change to "post-correction are necessary."

111) Lines 409-412. These sentences are all very vague. Please specify what "special caution" is needed. Please specify what is meant by "modernization". Which of these instruments are considered modern? Can you just say, "don't use the AE31" (or whichever is least appropriate)? What are the "means for field instruments' reference"? Should each site have a MAAP? Should MAAPs be shuttled around the Arctic to different sites? Can the COSMOS be a transfer standard since it's very stable and eliminates the scattering correction? Please be specific and very clear, since this is the "take-home" message the reader will get. Make sure this message is also in the abstract.

112) References. Not all the references are in Copernicus format. Some titles are capitalized, some are not. Some journals are abbreviated, some are not. This is a consequence of reference manager software. Please save the time of the technical editors by fixing everything before submission, since you'll have to do it anyway.

113) Figures 1, 2 and 4 will be very hard for anyone with a color impairment to read (~8% of European men). If you are using Igor, EOSSpectral16 is more friendly. In Figs. 1 and 2 you can distinguish curves using different symbols or line types in addition to making good color choices.

114) Fig. 4. Please make all fonts bigger. The figure caption should state "1-hour averaged absorption coefficient. . . . colored by 1-h averaged omega_0,635 nm. The solid lines are bivariate fits to the data, and the dashed lines are the 1:1 values. The corresponding. . . ."

---

## Author Response (AR2)

Dear Dr Charles Brock,

I am on behalf of all the authors of this manuscript very grateful for your excellent and prompt editing work that has truly helped to improve this manuscript. Also, I would like to express our thankfulness to both excellent reviewers who have done so great work. I appreciate this very much.

We truly hope that you find the manuscript improved, from the original and the previous version. Please, do not hesitate to contact me for any requests of modification or additional corrections, advises or technicalities.

Below you may find our answers to Reviewer #2 questions and concerns. The manuscript has been modified accordingly and marked in the pdf.

Kind regards,

Eija Asmi

Answers to Referee #2:

The authors have made some attempt to reframe the manuscript toward focusing on intercomparing the absorption instruments, which is an improvement. However, the overall depth of analysis for this paper remains very shallow, and I'm left wondering what the main findings / recommendations of this work are for the scientific community as promised by the last line of the abstract: "provide some useful guidelines for instruments selection and uncertainty analysis". While the revision appears to be moving in the right direction, the manuscript is not yet at a level of scientific quality where I could recommend it for publication in AMT. Consequently, I recommend that the manuscript be returned to the authors for another major revision/rewrite that focuses on improving its scientific contribution and depth of analysis, conclusions, and recommendations.

REPLY: We thank the reviewer for acknowledging the work done for improving the manuscript. We are happy to hear the direction has been correct. We intend to continue to improve the presentation of our results, with the goal to reach a good quality manuscript with clear conclusions and a scientific message.

General comments:

1) What is the new development, significant advance, or novel aspect of in situ aerosol measurements being communicated by this manuscript? Why are the results meaningful and important?

REPLY: While several previous intercomparisons of absorption measurement instruments in both in field and lab have been done, their reported results highlight the instrument performances at medium- to high-concentration ranges and are nearly neglecting the measurements done at around or just above the zero concentrations (e.g. Mölders and Edwin, 2018). We found only couple of previous studies that have contributed to this major knowledge gap. The following text was added to introduction:

"The co-located PSAP and CLAP instruments showed a good agreement in real-world Arctic inter-comparison (Ogren et al., 2017), but the study did not include Aethalometer or MAAP instruments, which are slightly different techniques. The co-located aethalometers in the Arctic showed relatively more discrepancies, which were discussed by Mölders and Edwin, 2018. However, to the best of our knowledge, comprehensive studies of co-located parallel filter-based instruments inter-comparisons in pristine field environments are lacking and that is one reason for the poorly quantified Arctic absorption baseline. "

We focus here to this previously neglected concentration range. We show using both new and previous results that at very pristine environments these low concentration levels are frequently observed, and form part of the important concentration baseline. According to the instrument manufacturers, the devices can measure down to very low concentrations. Here we present evidence that this is indeed quite the case, and estimate the lowest detection limit and accuracy of the absorption measurements.

Mölders, N. and Edwin, S. (2018) Review of Black Carbon in the Arctic—Origin, Measurement Methods, and Observations. *Open Journal of Air Pollution*, **7**, 181-213. doi: 10.4236/ojap.2018.72010.

Ogren, J.A., Wendell, J., Andrews, E. and Sheridan, P.J. (2017) Continuous Light Absorption Photometer for Long-Term Studies. Atmospheric Measurement Techniques, 10, 4805-4818. https://doi.org/10.5194/amt-10-4805-2017

2) Based on the instrument measurement intercomparison presented in Section 3.3 and in Figure 4, it is clear that there are significant differences between the absorption measurement techniques. In particular, the AE31 significantly underperforms the AE33, while even the AE33 and COSMOS appear to have non-zero intercepts. While the PSAP-MAAP comparison looks pretty good, there are also some notable departures from the 1:1 line where the PSAP is seeing very low or negative values. What is going on with the instruments and air masses during these periods? This intercomparison is the only real meat of this study, and there needs to be some additional discussion about what is being observed here and why (not just simply campaign-average slopes and R^2 values). In addition to adding more discussion to Section 3.3, I'd like to see recommendations for how to use these results to make better, future measurements of aerosol absorption, scattering, and extinction at low-aerosol, Arctic sites like Pallas.

REPLY: We thank reviewer for this excellent comment. Generally we feel that the filter-based absorption instruments agree surprisingly well in these demanding conditions, but as pointed out by reviewer, there are also several interesting discrepancies shown in Figures 2 and 4 that we examined more carefully. Most interestingly, at extremely high SSA values the PSAP scattering-corrected data appears to be "over-corrected" while at lower SSA values, the scattering-correction appears to be necessary to correct the data to a level of MAAP. In the manuscript the AE31 and the AE33 data were not corrected with a scattering-correction function due to the practical reason that it doesn't exist for AE33 and also, due to the reason that a common practice in GAW is to use Cref value instead. This difference finally shows in our data analysis giving support for the necessity of the scattering correction as a general rule, when possible, however, keeping in mind the possibly increasing noise. We added now more discussion on these matters in the text. In addition, we wrote our recommendations clearly in conclusions, including preference to select filter-based instruments in low-concentration environments, include sufficient averaging of data to reach the instruments detection limit, use of co-located instruments as transfer standards to estimate data precision, have co-located scattering measurements if not provided by the absorption instrument and to be

specifically cautious in data analysis at >0.99 SSA values, which still occur in the Arctic, as discussed in section "Representativeness of the measured absorption values".

3) A lot is made of the extinction minus scattering (EMS) absorption estimates, which one would've expected to perform poorly under low aerosol loadings and high scattering:extinction ratios (i.e., where one is differencing two relatively large numbers to calculate a much smaller absorption coefficient). True to these expectations, the differencing technique does perform poorly. I don't see this as a real meaningful or important finding and suggest that Figure 3 is not necessary.

REPLY: We have reduced discussion on EMS-technique and corresponding results as compared to the previous manuscript version. We still find it relevant to keep a small amount of results in the discussion, especially considering that EMS is a good candidate for traceable absorption reference method, it needs to be clearly acknowledged in the literature why it can not be used directly in the field, or at all concentration ranges that are met in the field, and is thus limited in range.

4) I understand the authors' hesitance to bring in new data and expand the scope of the study. Since there are previous long-term studies made a Pallas, it would be helpful to synthesize the results from those studies to contextualize the results being presented here. For example, what are the typical absorption, extinction, and scattering coefficients reported by these prior Pallas studies for June-July and throughout the year? How does the 2019 summer compare to "typical" conditions? Are there periods where aerosol absorption is higher than what was seen in 2019?

REPLY: We added a new subsection on this, putting these new finding into the context of Pallas (Arctic) "typical" absorption levels.

Specific Comments:

Line 125: What is ATN? What are the units of 60 ATN?

REPLY: ATN compares the light intensity after passing through a filter with aerosols to that of the intensity while passing through a clean filter. The definition is given e.g. on Backman et al. 2017, Eq. (1). Reference was added.

Line 142: What is insufficient about the manufacturer correction? This is not clear from this manuscript.

REPLY: The sentence is now corrected as: "…is too low for most atmospheric aerosols (Laing et al., 2020)".

Line 171-172: Please add some additional discussion about why the absorption Angstrom exponent would be uncertain for aerosol with SSAs close to unity. Is this because the absorption signals are low or are there scattering interferences that are not well accounted for? This uncertainty is likely to be especially important for the Pallas site.

REPLY: This comment makes reference to Backman et al., 2014 paper, where scatter in AAE is shown to increase with increasing SSA for Virkkula et al. 2010 correction (in particular). This was clarified in modified manuscript text: "However, it should be noted that at high SSA values the corrected a_AP becomes uncertain and should be interpreted with caution, as shown in Backman et al., 2014 Figure 7: intercomparison of correction algorithms.

Line 225: I don't understand why it is noted here that EMS is traceable to SI units. This is just differencing the extinction and scattering measurements discussed in previous sections.

REPLY: This is mentioned for the project connection to metrology.

Lines 226: It's true that differencing two measurements avoids filter-based artifacts, but there can also be other sources of uncertainty that are quite large. In particular, when the aerosol SSA ~ 1, the EMS technique becomes very uncertain as it involves difference two numbers of similarly large magnitude relative to the absorption.

REPLY: Exactly.

Line 239: How do the different geometry correction factors explain why the CAPSex and CAPSssa derived EMS calibrations would be different? Is this disagreement caused by the CAPS monitors or by the nephelometers?

REPLY: This was clearly confusingly written. The geometry correction factors were proposed to explain the differences observed in CAPS. There were very minor differences in nephelometers too, explained by other factors mentioned in text. The text was slightly improved.

Lines 239-241: I don't think that the instrument flow rates and/or inlet settings would have any impact on the correction factors. Hopefully, these calibrations were done with conductive tubing and at steady state conditions, and, ideally, with most of the actual ambient sampling setup. Are these sentences suggesting that there were significant particle transport losses that bias the calibration procedures?

REPLY: Agree. The calibrations were done with conducted tubing with similar length and setting, using an aerosol with Ångström exponent similar to outdoor air (in range 1.5-1.7). Text was slightly modified.

Line 286-287: I don't think it can be stated that there is even qualitative agreement. This is particularly the case in July where the EMS inferred values are trending up even as the direct measurements show no obvious trend.

REPLY: Good point. The agreement is weak, and in modified Fig. 2 it is obvious that EMS-derived absorption follows  more closely the scattering and extinction time series. Text was slightly modified to clarify this.

Line 288: Do you think that the high absorption values implied by EMS of ~ 3 Mm-1 are real or caused by taking the difference of two, large, noisy numbers?

REPLY: This is most likely the explanation, and was added in manuscript text.

Lines 312-313: I think it's fine to choose the MAAP as a common reference for comparison, but I'm confused about the rationale presented in the manuscript that there is "low demand of any artifact related post-correction in MAAP". What does this mean?

REPLY: Corrected as "MAAP is known to be essentially an artifact-free technique to measure aerosol absorption on a filter and has been widely utilized as a practical field reference method also in the past."

Lines 354-357: Please report averaging intervals for these limits of detection.

REPLY: Done.

Line 357: What support is there for the statement that EMS methods are applicable above 0.1 Mm-1? Indeed, I would think that the applicability of these methods would be dependent on both the instrument signal-to-noise as well as the aerosol SSA.

REPLY: This is true. We modified the sentence to state that our evidence only shows that the method is not applicable at below 0.1Mm-1 levels and that due to the nature of the method (taking a difference of two large numbers), it is supposedly also highly dependent on the aerosol SSA.

Figure 2: Please make these panels full page width and add dashed lines to the lower panel that show the -0.2 to 0.6 Mm-1 range. Also, please include additional panels showing the time series for the scattering and extinction coefficient measurements as well as the SP2 BC timeseries.

REPLY: These were added in Figure 2.

---

## Author Response (AR3)

Dear Editor,

Please find attached a modified version of the manuscript. We have corrected all those technical issues that you kindly pointed out in the manuscript text and in presentation technical quality. We have done final reading and check of the manuscript overall content and minor typos. We hope that this version could be adequate for publication in AMT.

We are very grateful for your excellent editing work; for all the time and effort, patience, and very good suggestions, all that has helped us significantly to improve the manuscript quality. Thank you so much.

With kind regards,

Eija Asmi on behalf of all the authors in the manuscript